# The dual life of disordered lysine-rich domains of snoRNPs in rRNA modification and nucleolar compaction

Carine Dominique[1,5], Nana Kadidia Maiga[1,5], Alfonso Méndez-Godoy [2], Benjamin Pillet [2], Hussein Hamze [1], Isabelle Léger-Silvestre[1], Yves Henry[1], Virginie Marchand [3], Valdir Gomes Neto [4], Christophe Dez[1], Yuri Motorin[3], Dieter Kressler [2]✉, Olivier Gadal [1]✉, Anthony K. Henras [1]✉ & Benjamin Albert [1]✉

Intrinsically disordered regions (IDRs) are highly enriched in the nucleolar proteome but their physiological role in ribosome assembly remains poorly understood. Our study reveals the functional plasticity of the extremely abundant lysine-rich IDRs of small nucleolar ribonucleoprotein particles (snoRNPs) from protists to mammalian cells. We show in *Saccharomyces cerevisiae* that the electrostatic properties of this lysine-rich IDR, the KKE/D domain, promote snoRNP accumulation in the vicinity of nascent rRNAs, facilitating their modification. Under stress conditions reducing the rate of ribosome assembly, they are essential for nucleolar compaction and sequestration of key early-acting ribosome biogenesis factors, including RNA polymerase I, owing to their self-interaction capacity in a latent, non-rRNA-associated state. We propose that such functional plasticity of these lysine-rich IDRs may represent an ancestral eukaryotic regulatory mechanism, explaining how nucleolar morphology is continuously adapted to rRNA production levels.

Ribosome assembly is an incredibly complex process orchestrated in time and space by a plethora of assembly and maturation factors (AMFs), which are required to convert large ribosomal RNA (rRNA) precursors and around 80 ribosomal proteins (RPs) into mature ribosomes on a time scale of minutes[1–3]. This process is initiated in a dedicated membrane-less organelle, the nucleolus, whose size, shape and morphology are highly dynamic depending on rRNA synthesis[4,5]. Importantly, the role of the nucleolus goes beyond its function in ribosome biogenesis as it also acts as a sensor to mediate stress adaptation through its ability to sequester/release specific factors according to growth conditions[6,7].

During exponential growth, the nucleolus is organized in three functionally distinct subdomains[8,9]. The fibrillar centers (FCs) are enriched in ribosomal DNA (rDNA) and RNA polymerase I (RNAPI), the dense fibrillar components (DFCs) contain small nucleolar ribonucleoprotein particles (snoRNPs) and early associating AMFs, and the granular component (GC) contains later pre-ribosomal particles. The earliest stages of ribosome biogenesis take place at the interface between FCs and DFCs with the synthesis of a large rRNA precursor (pre-rRNA). Two families of snoRNPs associate with nascent pre-rRNAs to introduce nucleotide modifications and assist their proper folding, two essential aspects of their maturation into the 18S, 5.8S and 25S rRNAs of functional ribosomes in yeast and human cells[10–12]. C/D-type snoRNPs are composed of box C/D snoRNAs, the methyltransferase Nop1/Fibrillarin and three additional core proteins Snu13, Nop56 and Nop58 (Fig. 1A); most of these particles introduce ribose methylations

[1]Molecular, Cellular and Developmental (MCD) Unit, Centre for Integrative Biology (CBI), CNRS, University of Toulouse, UPS, Toulouse, France. [2]Department of Biology, University of Fribourg, Fribourg, Switzerland. [3]CNRS-Université de Lorraine, UAR2008 IBSLor/UMR7365 IMoPA, Nancy, France. [4]Department of Biochemistry, Institute of Chemistry, University of São Paulo, São Paulo, Brazil. [5]These authors contributed equally: Carine Dominique, Nana Kadidia Maiga. ✉e-mail: dieter.kressler@unifr.ch; olivier.gadal@univ-tlse3.fr; anthony.henras@univ-tlse3.fr; benjamin.albert@univ-tlse3.fr

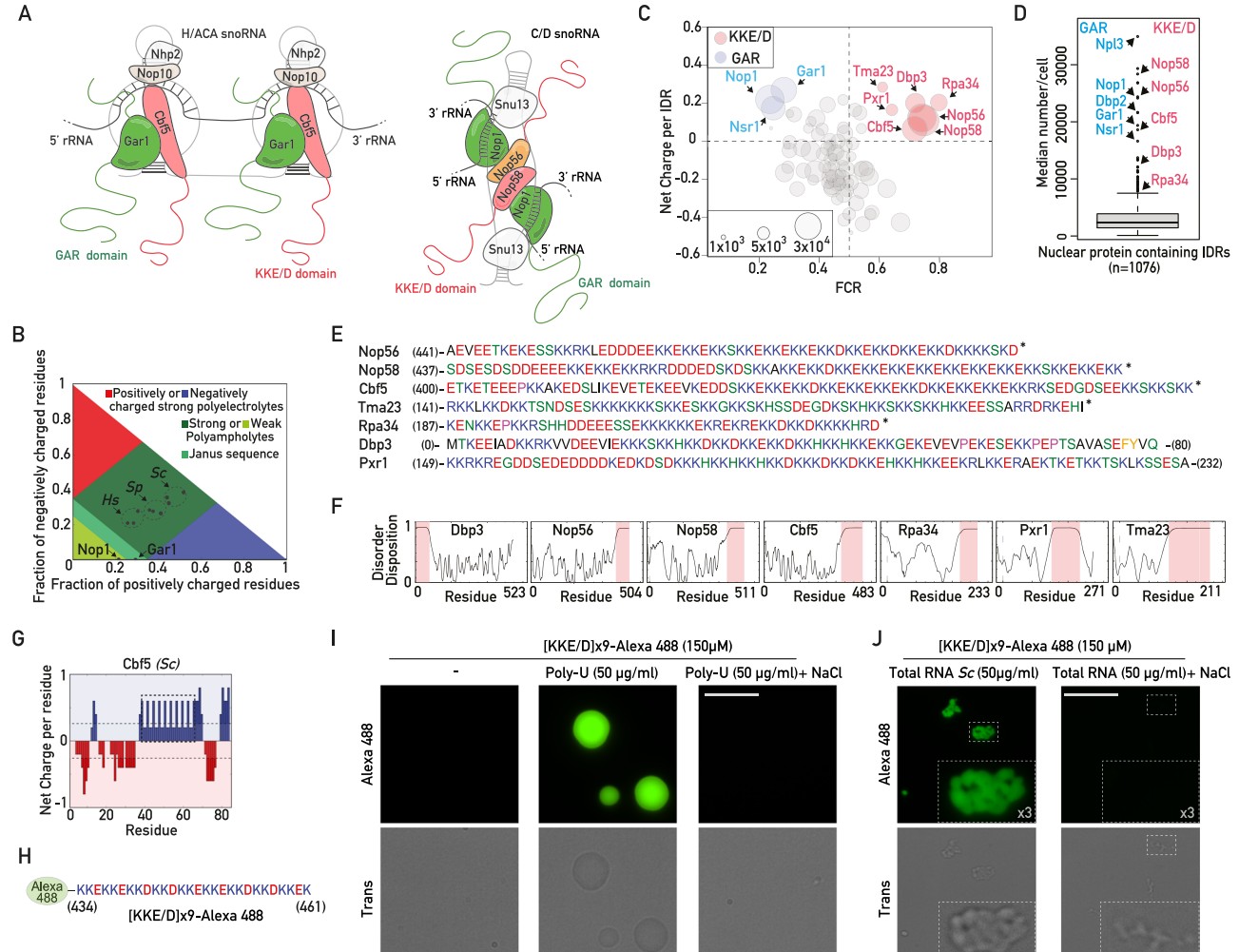

**Fig. 1 | A lysine-rich IDR is present in a subset of abundant, early-acting ribosome biogenesis factors. A** Schematic representation of H/ACA and C/D snoRNPs. The KKE/D and GAR domains are shown in pink and green, respectively. **B** Diagram of states for the KKE/D domains of *S. cerevisiae* (*Sc*) Nop56, Nop58, Cbf5 and their *H. sapiens* (*Hs*) or *S. pombe* (*Sp*) homologs. The position of the GAR domains of Nop1 and Gar1 is also indicated. The red and blue areas correspond to strong polyelectrolyte features with FCR > 0.35 and net charge per residue >0.3 exhibiting coil-like conformations. **C** Classification of *S. cerevisiae* IDRs longer than 30 amino acids (*n* = 110) according to their Frequency of Charged Residues (FCR) and Net Charge as defined in Holehouse et al.[65]. The size of each circle is proportional to protein abundance (Ho et al.[66]). Blue and red circles correspond to GAR domain- and KKE/D domain-containing proteins, respectively. **D** Box plot showing the abundance of all nuclear proteins (*n* = 1076) containing IDRs longer than 30 amino acids. Specific outliers containing KKE/D and GAR domains are indicated. **E** KKE/D domain amino

acid composition. The numbers correspond to the amino acid positions in the full-length sequence. *: C-terminus of the proteins. **F** Prediction of disorder tendency with PONDR[69]. **G** Net charge per residue distribution obtained using CIDER (http://pappulab.wustl.edu/CIDER/analysis/) in the KKE/D domain of *S. cerevisiae* Cbf5. The region highlighted by the dotted rectangle corresponds to the large block of lysine doublets. **H** Amino acid sequence of the peptide corresponding to the block of lysine doublets of Cbf5 highlighted in (**E**), which was coupled to Alexa488 ([KKE/D] x9-Alexa488). **I** Bright-field transmission (trans) and fluorescence (Alexa488) images of coacervate droplets formed in vitro by the [KKE/D]x9-Alexa488 peptide in the absence (-) or in the presence of poly-U RNAs and in the absence or presence of 300 mM NaCl. Scale bar = 15 µm. **J** As in (**I**) but coacervates were formed by mixing the [KKE/D]x9-Alexa488 peptide with total RNA from *S. cerevisiae* in the absence or presence of 300 mM NaCl. Scale bar = 15 µm. Zooms within the indicated dotted squares are shown. Source data are provided as a Source data file.

on selected rRNA nucleotides. H/ACA-type snoRNPs result from the association of box H/ACA snoRNAs with the pseudouridine synthase Cbf5/Dyskerin and the three core proteins Nop10, Nhp2 and Gar1 (Fig. 1A); they isomerize selected rRNA uridines into pseudouridines. In both cases, the snoRNA components of the snoRNPs base-pair with the pre-rRNA in the vicinity of the nucleotides to be modified, thereby guiding the enzymatic modifications catalyzed by Nop1/Fibrillarin or Cbf5/Dyskerin. In this process, the RNA helicases Dbp3 and Prp43, the latter in association with G-patch domain-containing co-factors Pxr1 and Tma23[13] (Méndez-Godoy and Kressler, manuscript in preparation), are crucial to regulate the dynamics of snoRNP association/dissociation[14,15]. However, it remains an enigma how the very early

recruitment of snoRNPs and RNA helicases to the nascent pre-rRNAs is achieved[11].

Interestingly, in silico analyses revealed that several protein components of human snoRNPs as well as a large number of nucleolar AMFs contain Intrinsically Disordered Regions (IDRs) in their N- or C-terminal extension[16,17]. A variety of roles have been attributed to IDRs due to their ability to establish transient and multivalent interactions[18]. Accordingly, they could function as targeting or docking signals, RNA/protein chaperones, or enable formation of high local concentrations of biomolecules in subcellular complexes termed condensates. The charge properties of IDRs could also be essential to regulate the flux of macromolecules

between coexisting nucleolar subdomains[16,19]. Importantly, the properties of IDRs are often conditioned by the solvent and molecular environment, which change the structural and interaction properties of these domains, allowing functional plasticity, i.e., a given IDR fulfills different functions depending on its local environment. Considering the dynamic structure of the nucleolus and the conditional properties of IDRs, the role of nucleolar IDRs during stress could be different from that fulfilled during optimal growth conditions. Therefore, it is essential to distinguish the roles of IDRs when the IDR-containing nucleolar proteins are engaged (active state) or not engaged (latent state) in pre-ribosomal particles[20]. Finally, the versatile properties of IDRs complicate the interpretation of the classical structure-function approaches and it is therefore of high importance to study their functional role(s) in the cellular environment and/or at least in the context of biologically relevant complexes. Recently, liquid-liquid phase separation (LLPS) has provided a possible mechanism of action of these nucleolar IDRs[4,21]. LLPS is driven by low-energy interactions between specific proteins containing IDRs, resulting in the compartmentalization of macromolecules into coexisting subdomains. For instance, self-interaction of the glycine/arginine-rich (GAR) domain, an abundant IDR associated to Fibrillarin (Nop1 in yeast), is essential to form a nucleolar microdomain in the DFC, allowing correct 5'ETS rRNA maturation[22]. Finally, it has been proposed that the multilayered architecture of the nucleolus arises through multiphase liquid immiscibility[4,21], but this area is subject to intense debate and controversy[20,23]. Alternative, LLPS-independent models have been proposed for the nucleolus, in which pre-rRNA production is sufficient to create both the vectorial flux and multilayered nucleolar organization by attracting AMFs that continuously oscillate between active and inactive states, i.e., the states in which AMFs are respectively associated or not with pre-rRNAs or pre-ribosomal particles[20]. Finally, although numerous nucleolar IDR-containing proteins form immiscible droplets in one- and two-component in vitro systems, mimicking the spatial partitioning observed in the crowded nucleolus is not trivial, and whether and how nucleolar IDRs contribute to nucleolar protein activity in the fluctuating context of ribosome biogenesis remain open questions.

In this study, we describe the functional plasticity of one of the most abundant lysine-rich nucleolar IDRs in budding yeast, present in snoRNPs and other early-acting AMFs. We delineate the molecular mechanisms underlying its function in rRNA modification and the regulation of nucleolar organization depending on rRNA production levels and the fluctuating pool of latent snoRNPs. Given the evolutionary conservation of large lysine-rich IDRs within snoRNPs in eukaryotic cells, we propose that this dual, growth-dependent role of lysine-rich IDRs may represent an ancestral regulatory system explaining how eukaryotic cells continuously adjust nucleolar morphology and rRNA production.

## Results

### An evolutionarily conserved lysine-rich IDR with unique features but unknown function is associated with eukaryotic snoRNPs

It has been previously reported that both H/ACA and C/D snoRNPs in mammalian cells contain two large IDRs[16,17], the glycine/arginine-rich (GAR) domain and a functionally uncharacterized lysine-rich IDR. The GAR domain has been extensively studied but the role of the lysine-rich domain remains unknown. Interestingly, we noted that long lysine-rich IDRs can also be identified in snoRNPs in the major branches of the eukaryotic lineage, including early diverging unicellular eukaryotes such as the protists *Euglena gracilis* or *Giardia intestinalis* (Supplementary Fig. 1A). They are absent from archaea, suggesting that lysine-rich IDRs are one of the major features of eukaryotic snoRNPs. In contrast to the GAR domains, the lysine-rich IDRs of yeast nucleolar proteins are

defined as strong polyampholytes in a diagram-of-states as defined previously[24] (Fig. 1B and Supplementary Fig. 1B–D). Strong polyampholytes carry both positive and negative charges along their backbone and their conformational behavior highly depends on the fraction of charged residues (FCR), net charge per residue (NCPR) and Kappa (K) index accounting for the charge segregation pattern of IDRs[24]. A Kappa index between 0.1 and 0.4, as observed for the IDRs of Cbf5 in eukaryotes (Supplementary Fig. 1B, C), predicts the ability of these IDRs to self-associate by electrostatic interactions between different blocks of positively and negatively charged domains. To get an overview of the properties of the lysine-rich IDR in comparison to other IDRs coexisting in the nucleolus, we classified all the IDRs of ribosome biogenesis factors in *S. cerevisiae* according to their fraction of charged residues, the net charge per domain (NCPD) and their abundance (Fig. 1C and Supplementary Data 1). As expected, the GAR domains of Nsr1 and H/ACA and C/D snoRNPs (carried by Gar1 for H/ACA snoRNPs or Nop1 for C/D snoRNPs) account for the most abundant positively charged IDRs (Fig. 1C). Importantly, GAR domains are also present in other abundant nuclear proteins, such as Npl3 or Dbp2 (Fig. 1D and Supplementary Fig. 1E), which are involved in mRNA metabolism[25,26], suggesting that the specific function and/or the nucleolar targeting of snoRNPs cannot depend solely on the GAR domain. We noted that the lysine-rich extensions of snoRNPs in yeast, hereafter called the KKE/D domain, belong to a small group of collectively extremely abundant IDRs (Fig. 1D) with unique electrostatic properties, i.e., a high FCR and a positive net charge (Fig. 1C, E). This specific lysine-rich IDR is localized at the N- or C-terminus of nucleolar proteins (Fig. 1F). Remarkably, the KKE/D domain is systematically associated with proteins involved in rRNA production and modification. Indeed, all snoRNPs contain two copies of the KKE/D domain (Cbf5 for H/ACA snoRNPs; Nop56 and Nop58 for C/D snoRNPs[27,28], Fig. 1A). It is also present in RNAPI (Rpa34 subunit)[29] and in RNA helicases or associated factors regulating the association/dissociation of snoRNPs: Dbp3[14], the G-patch protein Pxr1 regulating the activity of the RNA helicase Prp43[13] and the Pxr1-related protein Tma23 (Méndez-Godoy and Kressler, manuscript in preparation). KKE/D domains contain alternating blocks of positively and negatively charged residues including a specific region encompassing several lysine doublets spaced by polar amino acids, mostly glutamic or aspartic acids, but also less frequently histidine or serine residues (Fig. 1E, G). This large block of lysine doublets might participate in promoting intermolecular electrostatic interactions with the negatively charged phosphate backbone of nascent rRNAs. In this sense, we showed that fluorescent peptides corresponding to this specific domain formed coacervates in vitro in the presence of poly-uridine (poly-U) RNAs (Fig. 1H, I), indicating that these KKE/D repetitions contact single stranded RNAs. Formation of these coacervates relies on electrostatic interactions as the presence of 300 mM sodium chloride both prevented their formation (Fig. 1I) and dispersed these structures when added post-formation (Supplementary Fig. 1F). Condensate formation was not restricted to poly-U as in the presence of *S. cerevisiae* total RNAs, exhibiting more diverse and complex large RNA structures, KKE/D domains also formed multiple condensates (Fig. 1J). Interestingly, these condensates were also sensitive to the addition of sodium chloride as we could not visualize the accumulation of KKE/D peptides around these large RNA structures visible in the bright field microscopy channel in the presence of salt added either during or after formation of these condensates (Fig. 1J and Supplementary Fig. 1F). These observations in a simplified in vitro system are far from recapitulating the complexity of the nucleolar context in living cells. Nevertheless, the evolutionary conservation as well as the specific features of KKE/D domains determined in vitro and in silico

strongly suggest that lysine-rich IDRs might be key components of eukaryotic snoRNPs, for reasons that remain to be identified.

## snoRNP KKE/D domains are collectively required for growth and rRNA maturation

So far, KKE/D domains have been removed individually from the nucleolar proteins Cbf5, Nop56, Nop58, Pxr1 and Rpa34, which did not induce pronounced growth defects, destabilization of snoRNP particles or nucleolar localization defects[27,30–32]. We confirmed these previous observations (Fig. 2A and Supplementary Fig. 2A, B) but noted that the triple deletion of the KKE/D domains of both C/D and H/ACA snoRNPs (those of Cbf5, Nop56 and Nop58, $\Delta\Delta\Delta kk$) induced a significant growth delay at 30 °C (Fig. 2A and Supplementary Fig. 2A, B). A correlated mild early processing defect was revealed by the accumulation of the early 35S and 23S pre-rRNAs and the reduction in the levels of the 27SA$_2$ precursor to the large subunit rRNAs (Fig. 2B and Supplementary Fig. 2C–E). This growth defect was greatly enhanced at 37 °C (Fig. 2A and Supplementary Fig. 2A, B), or in cells additionally lacking the KKE/D domain of Rpa34, Pxr1 and Tma23 ($\Delta\Delta\Delta\Delta\Delta\Delta kk$, Fig. 2A); in line with this, a stronger accumulation of the 35S and 23S precursors could be observed in this sextuple mutant strain (Fig. 2B and Supplementary Fig. 2C–E). Moreover, the absence of multiple KKE/D domains also induced a higher sensitivity to BMH-21 (Fig. 2A and Supplementary Fig. 2A), a compound affecting specifically RNAPI activity[33], suggesting that KKE/D domains are particularly required when rRNA production decreases. These results indicate that KKE/D domains are collectively important for efficient growth and pre-rRNA processing.

## The KKE/D domain is required for efficient methylation and pseudouridylation of rRNAs

We further analyzed the rRNAs of mutant strains lacking KKE/D domains using RiboMeth-Seq (RMS) and HydraPsi-Seq (HPS) methodologies[34,35] to assess respectively the methylation of ribose sugars at the 2'-O position and the isomerisation of uridines into pseudouridines, which are thought to occur very rapidly on the nascent RNAPI transcripts. Interestingly, the absence of the KKE/D domain of Cbf5 induced a strong decrease in rRNA pseudouridylation (Fig. 2C, D and Supplementary Data 2), while ribose methylations remained unaffected (Fig. 2F and Supplementary Data 3). Conversely, analysis of rRNA 2'-O-ribose methylation and pseudouridylation levels in the absence of the KKE/D domains of C/D snoRNPs (those of Nop56 and Nop58) revealed that global rRNA methylation (Fig. 2E, F), but not pseudouridylation (Fig. 2D and Supplementary Data 2), was largely affected. We observed an interesting heterogeneity in the pseudouridylation or 2'-O-ribose methylation patterns (Fig. 2C, E), as some pseudouridylations and ribose modifications were strongly reduced or almost abolished, while others were not affected, suggesting that these site-specific rRNA modification defects cannot be explained by a global decrease in the catalytic activity of Cbf5 or Nop1. In line with this, it has been reported that the absence of Cbf5's KKE/D domain does not affect the pseudouridylation activity of in vitro reconstituted H/ACA snoRNPs[36]. Furthermore, in cells lacking the KKE/D domains of Pxr1 and Tma23, rRNA ribose methylation was also strongly affected in the first half of the 25S rRNA (Fig. 2G), which coincides with previously reported binding sites of the Prp43 RNA helicase[37]. These results demonstrate that the KKE/D domains of snoRNPs or of G-patch proteins functionally linked to Prp43 are required for efficient rRNA modification.

## The KKE/D domain is not required for nucleolar localization but is critical for accumulation of associated proteins in the vicinity of rDNA genes

The specific KKE/D domain features and the rRNA modification defects observed in the absence of KKE/D domains prompted us to test whether this IDR might be involved in the targeting to a subnucleolar environment in the vicinity of nascent rRNAs dedicated to the earliest steps of ribosome biogenesis. In agreement with previous reports[28,38], deletion of the KKE/D domain of Cbf5 (Cbf5-Δkk, Fig. 3A) did not prevent nucleolar targeting of C-terminally GFP-tagged Cbf5 (Fig. 3B and Supplementary Fig. 3A). However, fluorescence microscopy signal quantifications indicated that nucleolar accumulation of Cbf5 was weakly but significantly decreased in the absence of its KKE/D domain (Fig. 3C) and a nucleoplasmic signal became readily detectable (Fig. 3B and Supplementary Fig. 3A). In order to test whether KKE/D domains might promote accumulation in the vicinity of transcribing polymerases in the DFC, where snoRNPs and certain RNA helicases are expected to interact with the nascent, unfolded pre-rRNA, we assessed by chromatin immunoprecipitation (ChIP) the association of Cbf5 with rDNA genes. Strikingly, truncation of the KKE/D domain of Cbf5 drastically reduced its physical association with rDNA genes (Fig. 3D, E). This phenomenon was not restricted to Cbf5 since we observed that truncation of the KKE/D domain of Pxr1 had similar consequences (Supplementary Fig. 3B–E). We concluded that the KKE/D domain is not strictly required as a nucleolar localization signal (NoLS), but rather promotes efficient targeting of nucleolar proteins close to transcribed rDNA genes. Interestingly, a nucleolar morphology defect has been reported in strains expressing variants of Nop56 or Nop58 lacking their KKE/D domain[31], suggesting that, given the abundance of snoRNPs and their accumulation in the vicinity of rDNA in the DFC, a snoRNP targeting defect could have a direct impact on nucleolar organization. Remarkably, the absence of the KKE/D domains of snoRNPs ($\Delta\Delta\Delta kk$) completely abolished the visualization of DFCs, which can be identified by electron microscopy due to their typical contrast and fibrillar morphology (Fig. 3F). We confirmed this result using Correlative Light and Electron Microscopy (CLEM[39], allowing to control that the nuclear sections observed by electron microscopy indeed contain the nucleolar region (Net1-GFP fluorescent signal) in both wild-type and $\Delta\Delta\Delta kk$ cells (Supplementary Fig. 3F). In the absence of the KKE/D domains of C/D and H/ACA snoRNPs, the nucleolus appeared largely homogeneous (Fig. 3F and Supplementary Fig. 3F), indicating that the KKE/D domains are indeed critical for proper nucleolar organization.

## The KKE/D domain is sufficient to promote efficient targeting close to transcribed rDNA genes

We next analyzed the subcellular localization of a construct consisting in the C-terminal IDR of Cbf5 (Cbf5(392-483)) fused to GFP (KKE/D-GFP, Figs. 3A and 4A) in order to explore more specifically the intrinsic properties of KKE/D domains independently of snoRNP particles. In contrast to GFP alone, the KKE/D-GFP construct localized to the nucleus with a clear accumulation in the nucleolus (Fig. 4A and Supplementary Fig. 4A, B), suggesting that, although KKE/D domains are not essential for nucleolar targeting (Fig. 3B and ref. 27), they function as a NoLS on their own. Remarkably, the fusion of the KKE/D domain to GFP, even though decreasing the immunoprecipitation efficiency (Fig. 4B), led to a significantly more efficient immunoprecipitation of rDNA chromatin compared to the background levels obtained with the GFP control (Fig. 4C). Taken together with the drastic reduction in the ability of Cbf5 lacking its KKE/D domain, despite its nucleolar localization, to immunoprecipitate rDNA, the ability of KKE/D-GFP to be targeted to the nucleolus and to efficiently immunoprecipitate rDNA suggested that the KKE/D domain has an intrinsic property to target proteins to the neighborhood of RNAPI. In line with this, we showed that KKE/D-GFP was sufficient to immunoprecipitate RNAPI subunits under ChIP conditions (Fig. 4D). Furthermore, KKE/D-GFP was physically associated with the early rRNA precursors as assessed by native immunoprecipitation experiments (Fig. 4E), further supporting that the KKE/D IDR promotes targeting to the close

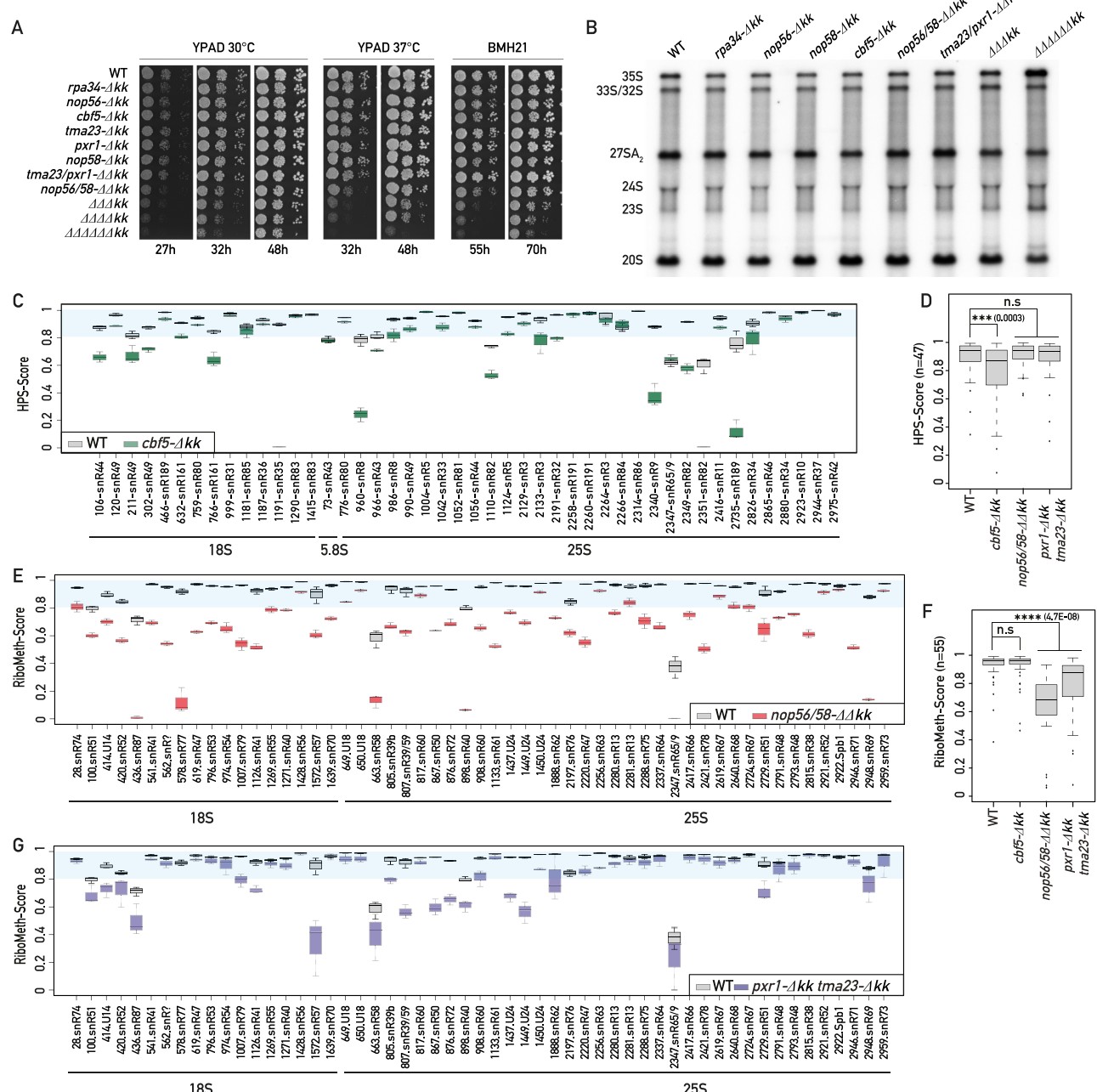

**Fig. 2 | snoRNP KKE/D domains are collectively required for optimal growth, pre-rRNA processing and rRNA modification. A** Tenfold serial dilutions of wild-type or mutant strains bearing individual or multiple deletions of KKE/D domains (-Δkk) were grown in YPAD medium for 27, 32 or 48 h at 30 °C or 37 °C or in the presence of a sub-lethal dose (15 μM) of BMH-21 for 55 or 70 h. *rpa34-Δkk* (Rpa34-(1-186)), *nop56-Δkk* (Nop56-(1-441)), *cbf5-Δkk* (Cbf5-(1-402)), *tma23-Δkk* (Tma23-(1-141)), *pxr1-Δkk* (Pxr1-(1-149)), *nop58-Δkk* (Nop58-(1-438)), *ΔΔΔkk (nop56-Δkk, nop58-Δkk, cbf5-Δkk), ΔΔΔΔkk (nop56-Δkk, nop58-Δkk, cbf5-Δkk, rpa34-Δkk), ΔΔΔΔΔΔkk (nop56-Δkk, nop58-Δkk, cbf5-Δkk, rpa34-Δkk, pxr1-Δkk, tma23-Δkk)*. *n* = 3 biologically independent experiments (Supplementary Fig. 2A). **B** Steady-state levels of rRNA precursors in the wild-type (WT) strain and the indicated KKE/D domain mutant strains. Total RNAs extracted from these strains were analyzed by Northern blotting using radiolabeled probes (23S.1 + 20S.3, Supplementary Data 8) detecting the indicated precursors. *n* = 4 biologically independent experiments (Supplementary Fig. 2C). **C**, **E**, **G** Comparison of the HydraPsi-Scores (HPS-Score) for each rRNA

pseudouridine in WT (gray) and *cbf5-Δkk* (green) strains (**C**). Comparison of the RiboMeth-Scores for each rRNA 2′-O-ribose methylation in WT (gray) and *nop56/58-ΔΔkk* (red) strains (**E**) or in WT (gray) and *pxr1-Δkk tma23-Δkk* (purple) strains (**G**). HydraPsi-Score and RiboMeth-Score indicate the fraction of uridine isomerisation and 2′-O-methylation at each site, respectively. The blue shaded area allows visualization of all sites that are highly modified (HPS or RMS scores >0.8). *n* = 3 biologically independent experiments. Box limits = 25th to 75th percentiles; line = median; Whiskers extend to 1.5 times the interquartile range on both ends. Source data are provided as a Source Data file. Box plots showing the HydraPsi-Scores (**D**) or RiboMeth-Scores (**F**) in wild-type (WT) and the indicated KKE/D mutant strains for all rRNA modification sites (*n* = 47 for HydraPsi-Scores and *n* = 55 for RiboMeth-Score). *p* values were calculated with unpaired two-samples Wilcoxon test. Significant differences are indicated by stars and with the exact *p* value on the graph. Source data are provided as a Source data file.

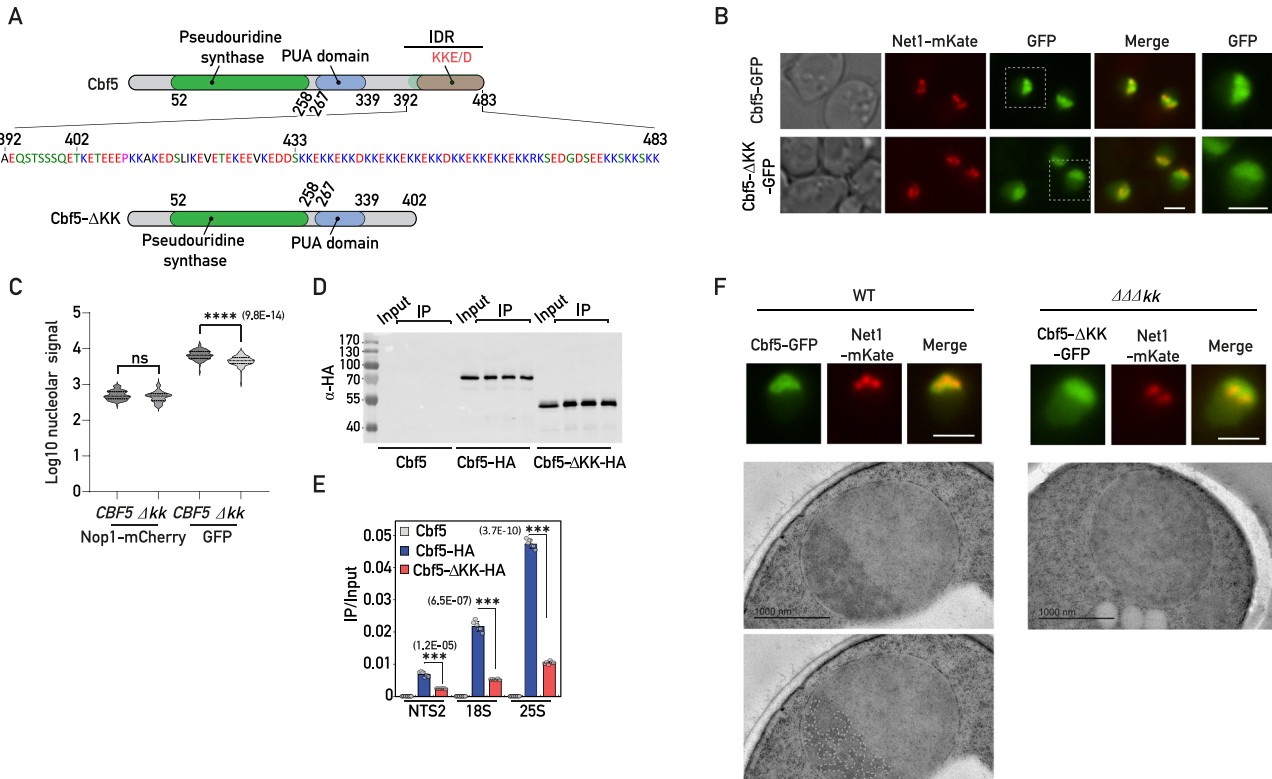

**Fig. 3 | The KKE/D domain is essential for recruitment to the vicinity of rDNA genes. A** Schematic representation of Cbf5 and Cbf5-ΔKK. The pseudouridine synthase, PUA and KKE/D domains are indicated in green, blue and red, respectively. The full IDR sequence (392-483) of Cbf5 is shown. The lysine-enriched region starts after amino acid 402 and the specific region containing several lysine doublets starts after amino acid 433. **B** *CBF5*-GFP and *cbf5-Δkk*-GFP strains expressing Net1-mKate, revealing the intranucleolar position of the rDNA, were grown exponentially and cells were analyzed by fluorescence microscopy. Merge: overlay of both fluorescent signals. Zooms of the GFP signals within the indicated dotted squares are shown on the right. Scale bars = 2 μm. **C** Quantification of the nucleolar Nop1-mCherry and GFP signals (Log10) in *CBF5*-GFP (*CBF5*; *n* = 89) and *cbf5-Δkk*-GFP (*Δkk*; *n* = 89) strains inspected in (**A**). *p* values were calculated with unpaired two-tailed Welch's *t* test. *n* = number of cells pooled from 3 biologically independent replicates. **D** Western blot analysis using anti-HA antibodies showing the IP efficiencies of Cbf5 (no tag control), Cbf5-HA or Cbf5-ΔKK-HA in the ChIP-

qPCR experiments shown in (**E**). Three technical IP replicates are shown for each condition as well as the corresponding input sample. **E** Cbf5 occupancy on rDNA genes at 18S, 25S or intergenic (NTS2) regions in strains expressing Cbf5 (no tag control), Cbf5-HA and Cbf5-ΔKK-HA evaluated by ChIP-qPCR. Immunoprecipitations were performed using anti-HA antibodies. Unpaired two-tailed t-test analysis was used for statistics. Data are presented as mean values ± SD. Significant differences are indicated by stars and with the *p* value on the graph. *n* = 6 biologically independent experiments. **F** Representative cells of *CBF5*-GFP or *cbf5-Δkk*-GFP, *nop56-Δkk*, *nop58-Δkk* (*ΔΔΔkk*) strains expressing Net1-mKate grown exponentially and analyzed as in (**B**) by fluorescence microscopy (top panels; scale bars = 2 μm) or for ultrastructural studies by transmission electron microscopy (bottom panels; scale bars = 1 μm). Position of the DFC in the wild-type nucleolus (WT) was determined by visual inspection of highly contrasted regions using ImageJ software before manual segmentation. A similar contrasted region is not observable in the *ΔΔΔkk* strain. Source data are provided as a Source data file.

proximity of transcribing RNAPI and mediates the association with nascent pre-ribosomal particles. As an alternative means of testing the in vivo targeting and interaction properties of KKE/D domains, we used a TurboID-based proximity labeling approach. To this end, we fused the KKE/D domain of Cbf5 (residues 392-483) to the improved TurboID biotin ligase (denoted BirA-KKE)[40] and used a TurboID-GFP fusion protein bearing the SV40-NLS at its N-terminus (NLS-BirA-GFP) as a background control for nuclear biotinylation (Fig. 4F). We next expressed separately these fusion proteins from a plasmid under the control of an inducible promoter in a wild-type strain (Fig. 4F and Materials and methods). As expected, BirA-KKE biotinylated a substantial number of nucleolar proteins (Fig. 4G and Supplementary Data 4, 5). Remarkably, however, we observed that biotinylation of the nucleolar proteins reported to be intimately associated with RNAPI[41] was especially efficient (Fig. 4H) and, in particular, that of all KKE/D domain-containing proteins (Fig. 4G). Our data demonstrate that the KKE/D domain shows an intrinsic property to target snoRNPs or other associated proteins to a dedicated sub-nucleolar compartment in the neighborhood of RNAPI.

## Efficient targeting of the snoRNP KKE/D domains to transcribed rDNA genes depends on rRNA production

Our data suggest that the targeting properties of the KKE/D domains could rely on direct interactions with nascent rRNAs. However, we could not exclude that direct interactions with RNAPI contribute to snoRNP recruitment. In order to test this, we used a yeast genetic background in which rRNAs are exclusively produced by RNAPII[42,43]. Remarkably, and contrary to the rRNA modification phenotypes observed in *cbf5-Δkk* and *nop56/58-ΔΔkk* cells, substitution of RNAPI by RNAPII had no significant effect on both rRNA ribose methylation and pseudouridylation (Supplementary Fig. 4C and Supplementary Data 2, 3). These results suggest that H/ACA and C/D snoRNPs are efficiently recruited to the vicinity of rDNA genes transcribed by RNAPII, strongly implying that rRNA precursor production per se, but not RNAPI, mediates efficient snoRNP recruitment. To assess this further, we tested the association of the KKE/D-GFP fusion protein with rDNA chromatin in the absence of RNAPI transcription, using the *rrn3-8* strain expressing a temperature-sensitive version of the RNAPI transcription initiation factor Rrn3. Upon RNAPI inhibition, KKE/D-GFP recruitment to rDNA was impaired (Fig. 4I),

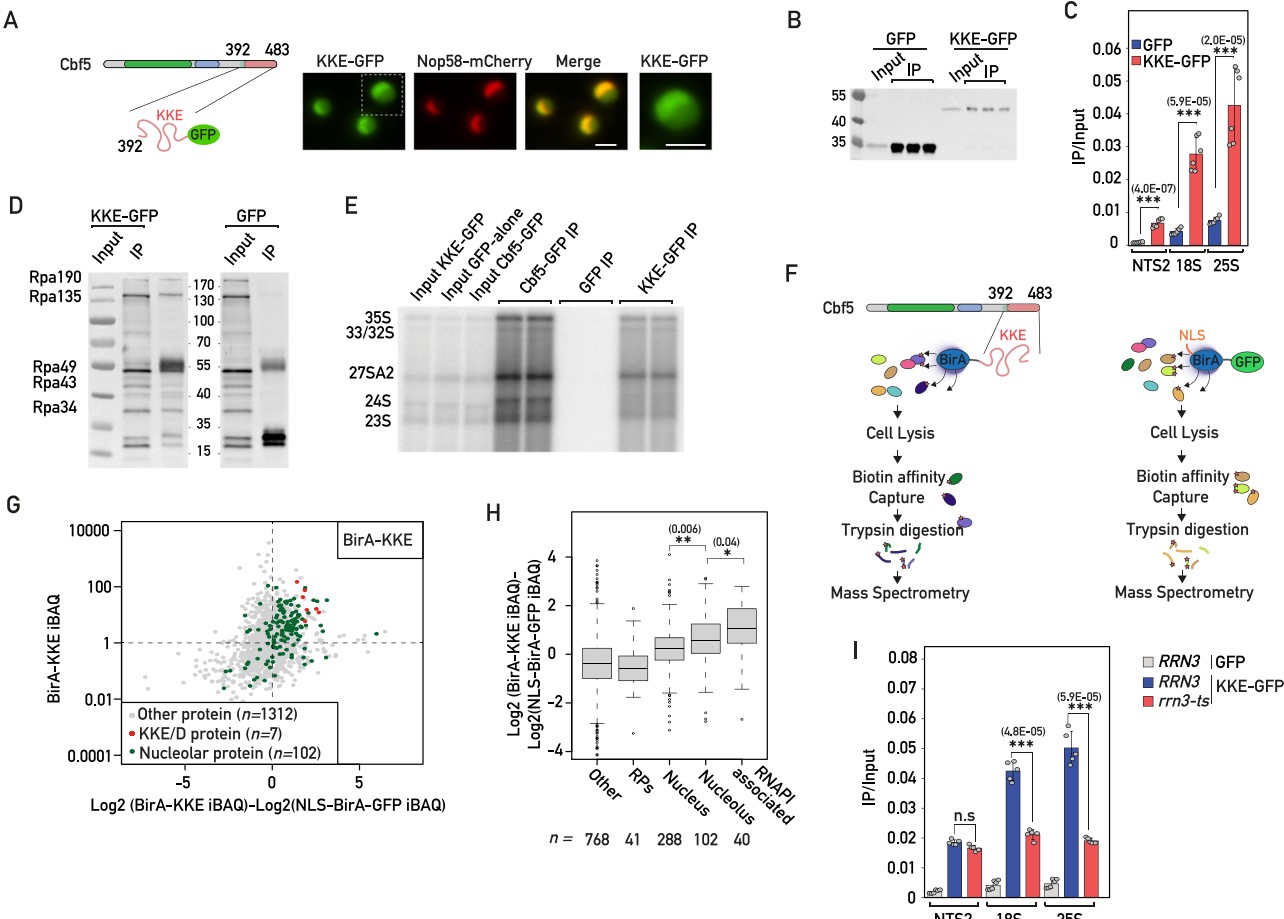

**Fig. 4 | The KKE/D domain is sufficient to promote efficient targeting close to transcribed rDNA genes. A** Strains expressing Nop58-mCherry and the KKE/D-GFP (KKE-GFP) construct were grown exponentially, and cells were analyzed by fluorescence microscopy. Zooms of the GFP signals are shown. Scale bars = 2 μm. **B** Western blot analysis using anti-GFP antibodies showing the IP efficiencies of GFP or KKE/D-GFP (KKE-GFP) in the ChIP-qPCR experiments shown in (**C**). **C** GFP and KKE/D-GFP (KKE-GFP) occupancy on rDNA genes at 18S, 25S or intergenic (NTS2) regions in wild-type cells evaluated by ChIP-qPCR. Unpaired two-tailed *t*-test analysis was used for statistics. Data are presented as mean values ± SD. Significant differences are indicated by stars and with the *p* value. *n* = 5 biologically independent experiment. **D** KKE/D-GFP (KKE-GFP) or GFP were immunoprecipitated using anti-GFP antibodies from formaldehyde-treated cells. Anti-RNAPI antibodies detecting all subunits were used to assess the co-immunoprecipitation of RNAPI. *n* = 2 biologically independent experiments. **E** KKE/D-GFP (KKE-GFP), Cbf5-GFP or GFP were immunoprecipitated using anti-GFP antibodies under native conditions. The co-immunoprecipitation of rRNA precursors was assessed by Northern blotting using radiolabeled probe 23S.1 (Supplementary Data 8). *n* = 2 biologically independent experiments. **F** Schematic representation of the TurboID-based

proximity labeling experiments. **G** Scatter plot showing the normalized abundance value (iBAQ, intensity-based absolute quantification) of each protein detected in the purification of biotinylated proteins from cells expressing the BirA-KKE bait plotted against the relative abundance of these proteins (log2-transformed enrichment) compared to their normalized iBAQ value in the control purification from cells expressing the NLS-BirA-GFP bait. KKE/D domain-containing proteins and nucleolar proteins are labeled. *n* is indicated for each category. Source data are provided as a Source Data file. **H** The proteins detected in the TurboID-based proximity labeling assay were classified into non-overlapping subsets including RPs, proteins associated with RNAPI, proteins reported to localize in the nucleus, in the nucleolus or in other cellular areas. The log2-transformed enrichment is given for each category. *p* values were calculated with unpaired two-tailed Welch's *t* test. Significant differences are indicated by stars and with the *p* value. *n* is indicated for each category. **I** KKE/D-GFP (KKE-GFP) or GFP occupancy on rDNA genes at 18S, 25S or intergenic (NTS2) regions in wild-type (*RRN3*) or *rrn3-8* (*rrn3-ts*) mutant cells grown at 37 °C. *p* values were calculated and indicated as in (**C**). *n* = 5 biologically independent experiments. Source data are provided as a Source data file.

demonstrating that KKE/D domain recruitment close to rDNA genes depends directly on rRNA production.

## KKE/D domains interact in a homo- and heterotypic manner

Taken together, our data support a model in which the lysine-rich IDRs of snoRNPs and other factors regulating snoRNP dynamics (Pxr1, Tma23, Dbp3) promote the targeting in close contact to nascent rRNAs. Interestingly, an unbiased yeast two-hybrid (Y2H) screen using the KKE/D domain-containing protein Rpa34 resulted in the isolation of few library clones encoding the KKE/D domain-containing proteins Nop56, Nop58 and Pxr1[44], suggesting that KKE/D domains may establish direct interactions. We thus directly tested using Y2H assays the interaction properties of KKE/D domains, both

the self-interaction of a given KKE/D domain (homotypic interactions) and the interaction between different KKE/D domains (heterotypic interactions). We observed a strong heterotypic interaction between the KKE/D domain of Rpa34 (residues 183-233) and those of Nop58 (residues 451-511) and Pxr1 (residues 172-213), while Nop56's KKE/D domain (residues 441-504) also interacted, albeit to a much lesser extent (Fig. 5A, B). Further Y2H assays revealed that full-length Pxr1 interacts with its own isolated KKE/D domain (homotypic interaction) and with those of Cbf5 (residues 433-483), Nop56 and Nop58 (Fig. 5C). In all cases, the KKE/D domain was both sufficient and required for the interactions (Fig. 5C). Moreover, the interaction of full-length Pxr1 with either Nop56 or Nop58 was dependent on their KKE/D domain (Supplementary

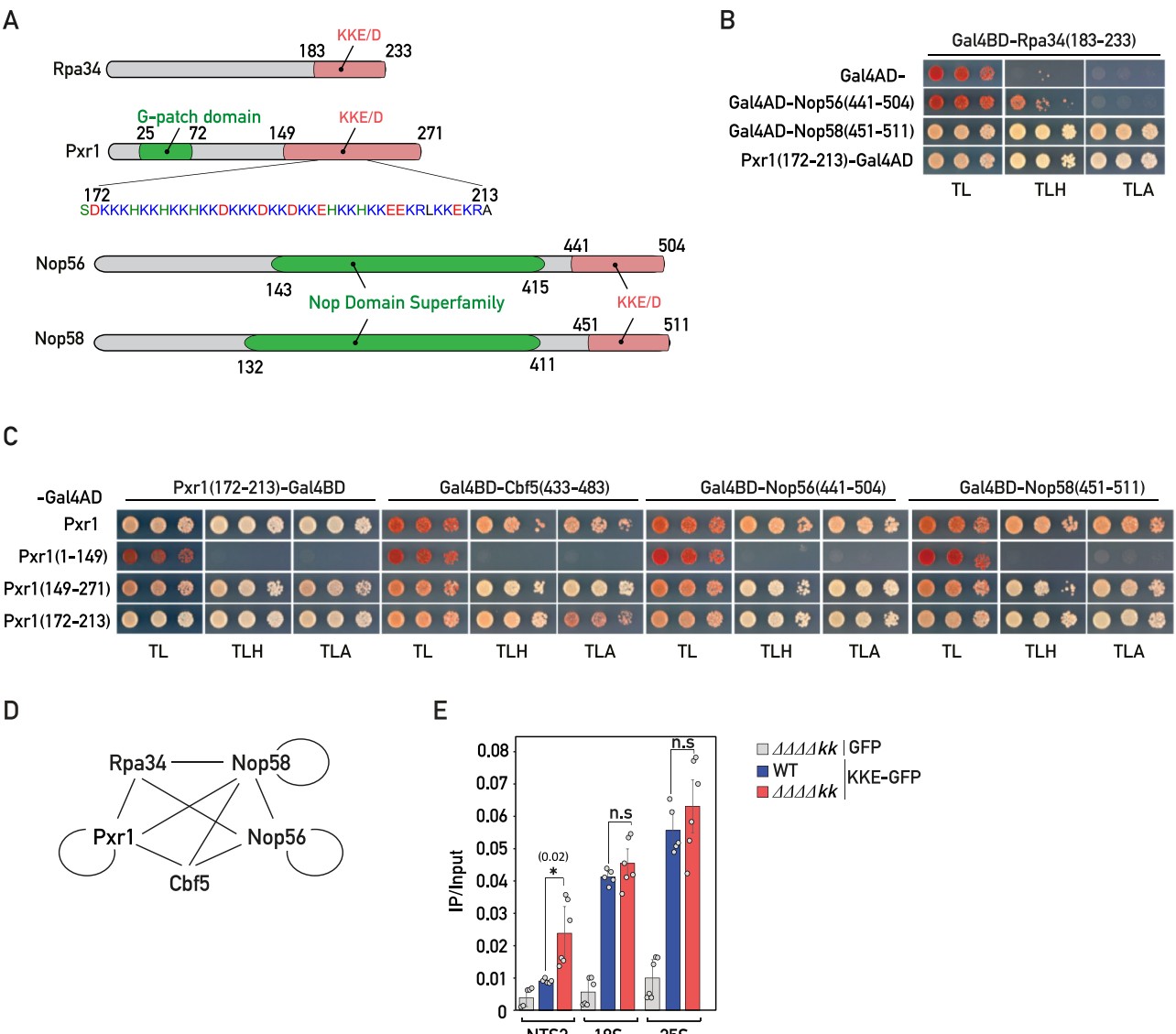

**Fig. 5 | KKE/D domains interact in a homo- and heterotypic manner.**
**A** Schematic representation of the domain organization of Rpa34, Pxr1, Nop56 and Nop58 proteins. **B** Yeast two-hybrid (Y2H) assays using the indicated combinations of the Gal4 activation domain (Gal4AD) alone or fused to the KKE/D domain of Nop56 (Nop56(441-504)), the KKE/D domain of Nop58 (Nop58(451-511)), or the KKE/D repeats of Pxr1 (Pxr1(172-213)) and the Gal4 DNA-binding domain (Gal4BD) fused to the KKE/D domain of Rpa34 (Rpa34(183-233)). Growth on SDC lacking tryptophan and leucine (TL) allowed to select cells containing both constructs; growth on SDC lacking tryptophan, leucine and histidine (TLH) indicated that the constructs interact; growth on SDC lacking tryptophan, leucine and adenine (TLA) indicated that the constructs interact strongly. *n* = 3 biologically independent experiments. **C** Y2H assays using the indicated combinations of the Gal4 activation domain (Gal4AD) fused to full-length Pxr1 (Pxr1), Pxr1 lacking its KKE/D domain (Pxr1(1-149)), Pxr1's isolated KKE/D domain (Pxr1(149-271)) or Pxr1's KKE/D repeats (Pxr1(172-213)) and the Gal4 DNA-binding domain (Gal4BD) fused to the KKE/D repeats of Pxr1 (Pxr1(172-213)), the KKE/D repeats of Cbf5 (Cbf5(433-483)), the KKE/D domain of Nop56 (Nop56(441-504)) or the KKE/D domain of Nop58 (Nop58(451-511)). Same legend as in (**B**). *n* = 3 biologically independent experiments. **D** Schematic representation of the interactions detected by Y2H assays between the KKE/D domains of Rpa34, Cbf5, Pxr1, Nop56 and Nop58 proteins. Circles correspond to self-interaction in a KKE/D domain-dependent manner. **E** KKE/D-GFP (KKE-GFP) or GFP occupancy on rDNA genes at 18S, 25S or intergenic (NTS2) regions in wild-type (WT) or *rpa34-Δkk, nop56-Δkk, nop58-Δkk, cbf5-Δkk (ΔΔΔΔkk)* quadruple mutant cells. GFP and KKE/D-GFP were immunoprecipitated using anti-GFP antibodies. DNA occupancy was defined as the ratio between the immuno-precipitation (IP) and the input signals. Two-tailed *t*-test analysis was used for statistics. Data are presented as mean values ± SD. Significant differences are indicated by stars and with the *p* value. *n* = 6 biologically independent experiments. Source data are provided as a Source data file.

Fig. 5A). Finally, we observed that homotypic interactions are not restricted to the IDR of Pxr1, as the KKE/D domain of Nop58 and, to a lesser extent, the one of Nop56 also exhibited self-interaction (Supplementary Fig. 5B). The propensity of KKE/D domain-containing proteins to interact with each other (Fig. 5D) could allow Rpa34 on RNAPI to nucleate the recruitment of early maturation factors such as snoRNPs. Nevertheless, as previously observed in cells in which rRNA is exclusively produced by RNAPII, we could not detect any significant change in rRNA modification in

*rpa34-Δkk* cells (Supplementary Fig. 4C and Supplementary Data 2, 3). Moreover, the fact that only rRNA pseudouridylation or methylation is affected in *cbf5-Δkk* or *nop56/58-ΔΔkk* cells, respectively, suggested that there is no high cooperativity between KKE/D domains for their recruitment to the nascent pre-rRNAs. In line with this, we demonstrated that rDNA association of the KKE/D-GFP construct, determined by ChIP, was not affected in the quadruple *cbf5-Δkk, nop56/58-ΔΔkk, rpa34-Δkk* mutant strain, in which about 70% of the KKE/D domains are missing (Fig. 5E). In conclusion, our

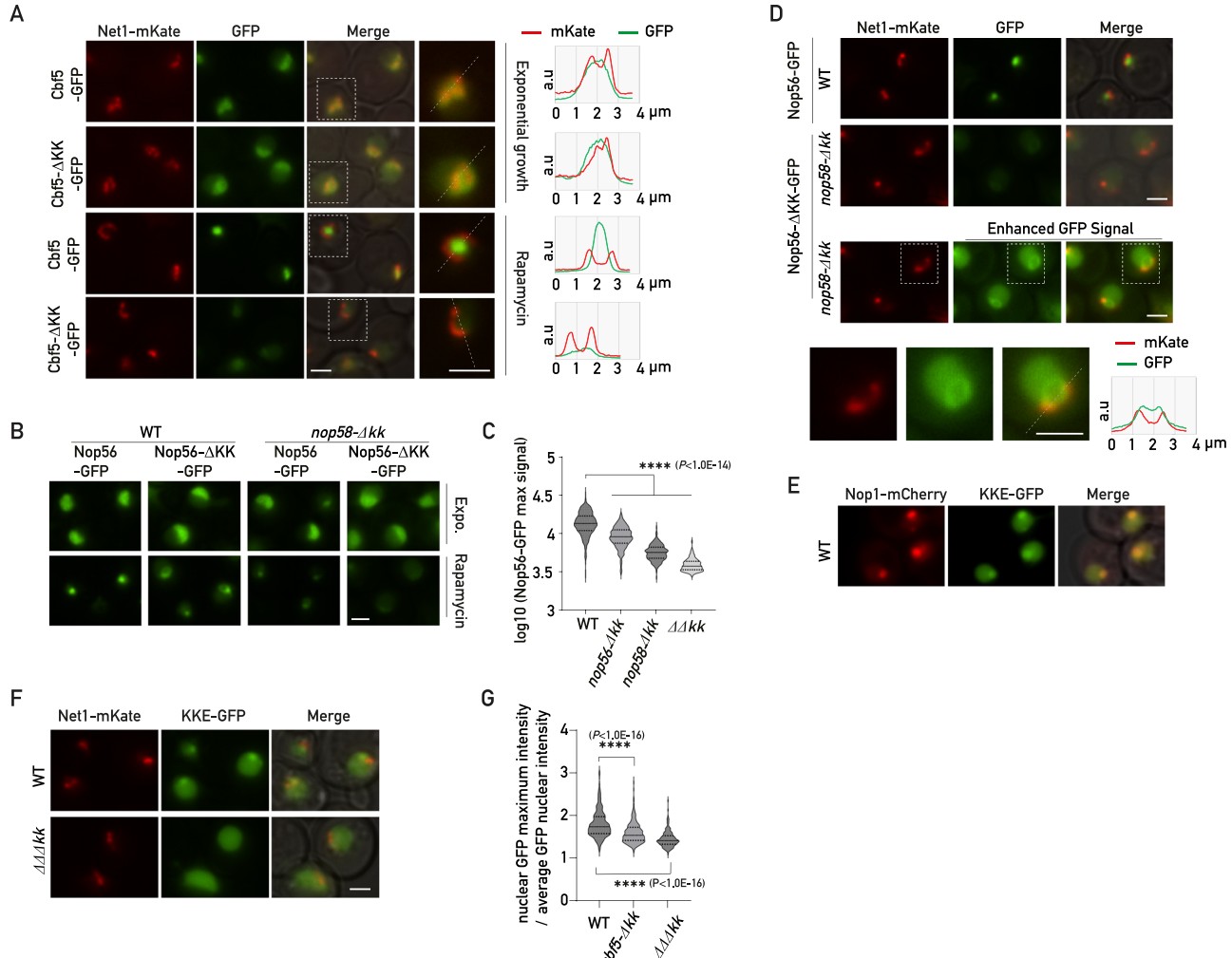

**Fig. 6 | KKE/D domains are essential for nucleolar compaction and sequestration of associated factors in a specific subnucleolar area following TORC1 inactivation. A** Fluorescence microscopy analyses of *CBF5*-GFP and *cbf5-Δkk*-GFP cells expressing Net1-mKate grown exponentially (Exponential growth) or treated for 4 h with rapamycin (Rapamycin). Zooms of the indicated dotted square areas are shown on the right and average plots (right panel) show the signal profiles of mKate and GFP in the nucleolus along the indicated dotted lines. Scale bars = 2 µm. **B** Fluorescence microscopy analyses of wild-type (WT) or *nop58-Δkk* cells expressing Nop56-GFP or Nop56-ΔKK-GFP grown exponentially (Expo.) or treated for 4 h with rapamycin (Rapamycin). Scale bar = 2 µm. **C** Quantification of maximum nuclear GFP signals (Log10) of wild-type or *nop58-Δkk* cells inspected in (**B**) expressing Nop56-GFP (WT; *n* = 149 and *nop58-Δkk*; *n* = 149) or Nop56-ΔKK-GFP (WT; *n* = 149 and *ΔΔkk*; *n* = 112) treated for 4 h with rapamycin. *p* values were calculated with unpaired two-tailed Welch's *t* test. *n* = number of cells pooled from 3 biologically independent replicates. **D** Fluorescence microscopy analyses of wild-type (WT) or *nop58-Δkk* cells expressing Nop56-GFP or Nop56-ΔKK-GFP or treated

for 4 h with rapamycin. Net1-mKate allows localization of rDNA in the vicinity of the Nop56 condensate. Merge: overlay of both fluorescent signals. Bottom panel: 16-bit brightness levels have been increased to specifically visualize the GFP signal in *nop56-Δkk*-GFP *nop58-Δkk* cells in the vicinity of the Net1-mKate signal. Zooms of the indicated dotted square areas are shown below. Right panel: GFP and mKate average plot signals. Scale bars = 2 µm. **E** Wild-type (WT) cells expressing Nop1-mCherry and the KKE/D domain of Cbf5 fused to GFP (KKE-GFP) were analyzed by fluorescence microscopy. **F** Same legend as in (**E**) for rapamycin-treated wild-type (WT) or *cbf5-Δkk nop56/58-ΔΔkk (ΔΔΔkk)* cells expressing Net1-mKate and KKE/D-GFP (KKE-GFP). Scale bar = 2 µm. **G** Quantification of maximum nuclear GFP signals (maximum intensity divided by average intensity) in rapamycin-treated wild-type (WT; *n* = 373), *cbf5-Δkk* (*n* = 249) or *cbf5-Δkk nop56/58-ΔΔkk (ΔΔΔkk; n* = 161) cells expressing the KKE/D-GFP construct. *p-values* were calculated and indicated as in (**C**). *n* = number of cells pooled from 3 biologically independent replicates. Source data are provided as a Source data file.

---

data do not support the hypothesis that interactions between KKE/D domains play a critical role in the targeting of snoRNPs and other factors to the nascent rRNAs.

## Cooperative interactions between KKE/D domains allow nucleolar compaction and sequestration of KKE/D domain-containing proteins under stress conditions

Our data revealed that the KKE/D domain interacts with RNA and mediates the recruitment of snoRNPs at the vicinity of the nascent rRNAs independently of other KKE/D domains. Paradoxically, we also showed that KKE/D domains mediate the self-interaction with other KKE/D domains as assessed by Y2H. Importantly, Y2H assays revealed

the interaction properties at the RNAPII promoters (*GAL1* and *GAL2*, respectively) of the *HIS3* and *ADE2* reporter genes, which are not embedded in the nucleolus in contact with nascent rRNAs. This latter observation suggests that the KKE/D domain self-interaction may operate in the absence of rRNA production or when the ribosome biogenesis rate decreases. We thus decided to explore using fluorescence microscopy the function of the KKE/D domains in the nucleolar environment under stress conditions reducing ribosome biogenesis, which increase the proportion of latent snoRNPs, i.e., snoRNPs not associated with nascent rRNAs. Under stress conditions reducing ribosome biogenesis activity, the nucleolus is reorganized into a compact structure, in which certain early factors, such as snoRNPs or

Nsr1, are concentrated at the center of the nucleolus, while other early (components of UTP complexes, U3-associated proteins) or late factors localize to outer layers through mechanisms that are not understood[45–47]. Using fluorescence microscopy, we first assessed the localization of Cbf5 following rapamycin treatment (Fig. 6A and Supplementary Fig. 6A, B), a drug mimicking nitrogen or carbon source exhaustion by inhibiting the growth regulator TORC1[48], or in the post-diauxic growth phase (Supplementary Fig. 6C). As expected, ribosome biogenesis inhibition triggered formation of a Cbf5 condensate adjacent to rDNA (Fig. 6A and Supplementary Fig. 6B), which also co-localized with Nop1-mCherry (Supplementary Fig. 6C). Remarkably, in the absence of its KKE/D domain, Cbf5 was no longer concentrated in this subnucleolar structure as only a small pool remained adjacent to rDNA genes (Fig. 6A and Supplementary Fig. 6A). Importantly, the reduced condensation of Cbf5-ΔKK-GFP was not linked to a rapid decrease of protein levels as Cbf5 and two other KKE/D domain-containing proteins Rpa34 and Nop56, with or without their KKE/D domain, remained stable following rapamycin treatment (Supplementary Fig. 6D). Similarly, deletion of both KKE/D domains of C/D snoRNPs totally abolished Nop56's condensation upon rapamycin treatment (Fig. 6B, C). Interestingly, in this double mutant, we observed that Nop56-ΔKK-GFP was even depleted from a specific area at the center of the nucleolus likely where snoRNPs equipped with KKE/D domains are normally concentrated (Fig. 6D and Supplementary Fig. 6E). We also showed that Pxr1-ΔKK-GFP did not accumulate together with snoRNPs in the absence of its KKE/D domain (Supplementary Fig. 6F, G). A similar phenomenon was observed for RNAPI (Rpa190-GFP) in the absence of Rpa34's KKE/D domain upon rapamycin treatment (Supplementary Fig. 6H, I). These data indicated that KKE/D domains provide a preferential access to RNAPI and other KKE/D domain-containing factors within a specific subnucleolar area when the rate of ribosome biogenesis is reduced. In line with this, a KKE/D domain alone, as shown here for the one of Cbf5, was sufficient to promote GFP accumulation in this specific subnucleolar structure upon rapamycin treatment (Fig. 6E, F). In order to test whether recruitment of the KKE/D-GFP construct depended on cooperative interactions with other snoRNP KKE/D domains, we assessed its localization in the absence of the KKE/D domain of H/ACA snoRNPs (cbf5-Δkk) or both H/ACA and C/D snoRNPs (ΔΔΔkk) (Fig. 6F, G). In line with a cooperative recruitment, nucleolar KKE/D-GFP foci formation was reduced or abolished in the absence of the snoRNP KKE/D domains. This result demonstrates that recruitment of KKE/D-GFP to this specific subnucleolar structure requires cooperative interactions between the KKE/D domains of snoRNPs. Importantly, this result also indicates that KKE/D-GFP alone is unable to form these subnucleolar structures, as we did not observe any condensate formation following expression of KKE/D-GFP in the absence of the KKE/D domains of snoRNPs (ΔΔΔkk, Fig. 6F). We next confirmed that formation of these subnucleolar structures upon inhibition of RNAPI transcription are directly linked to KKE/D-domain properties and cannot be recapitulated by any IDR with similar size. We either mutated the KKE/D domain of Cbf5 or replaced it by the polyampholyte IDRs of the nucleolar protein Enp1 (Supplementary Fig. 7A). These domains fused to Cbf5 are predicted in silico to be disordered (Supplementary Fig. 7B) but, as expected, they show different properties as illustrated on a phase diagram (Supplementary Fig. 7C). The mutant Cbf5 proteins are expressed at similar levels as the wild-type protein (Supplementary Fig. 7D) but trigger a slow-growth phenotype (Supplementary Fig. 7E) and affect the nucleolar targeting of Cbf5 in exponential growth condition (Supplementary Fig. 7F). Lastly, in contrast to wild-type Cbf5, these mutants lost their ability to form condensates in the nucleolus following stress (Supplementary Fig. 7F, G). In agreement with the strong conservation of lysine blocks in the IDRs of Cbf5 in the eukaryotic lineage, our data demonstrate that the KKE/D domain has specific properties essential for nucleolar organization.

## The snoRNP KKE/D domains act as ligands essential for regulating Nop1 and Gar1 condensation and multilayered nucleolar organization

Our data indicated that the KKE/D domains of snoRNPs are essential for the sequestration of other KKE/D domain-containing proteins. However, the KKE/D-GFP construct expressed alone was not sufficient to promote condensation, suggesting that other components participate in the formation of this subnucleolar structure in which snoRNPs are concentrated. It has been reported in numerous studies that the GAR domain-containing proteins of snoRNPs, Nop1 and Gar1, drive the assembly of phase separation droplets when they reach a sufficient concentration[21,49]. The ability of KKE/D domains to self-interact suggests that they could act as ligands between latent snoRNPs contributing to increase the local concentration of GAR domains (Fig. 7A) upon inhibition of ribosome biogenesis. In this sense, it has been proposed that protein condensation by phase-separation observed in vitro requires specific ligand proteins in vivo to locally concentrate IDRs prone to phase separation[50]. In line with this, we showed that both Gar1-GFP and Nop1-mCherry were not able to condensate in the absence of the KKE/D domains of H/ACA and C/D snoRNPs, respectively (Fig. 7B, C). Importantly, this sequestration defect was not common to any nucleolar protein as Enp1, a nucleolar AMF component of 90S pre-ribosomal particles[51], remained nucleolar in cells lacking all KKE/D domains of snoRNPs (Fig. 7D and Supplementary Fig. 8A, B). Nevertheless, we observed that the shape of the Enp1-GFP signal was more spherical, presumably due to the absence of the specific subnucleolar area where snoRNPs accumulate and from which Enp1-GFP is depleted (Fig. 7D and Supplementary Fig. 8C). In order, to obtain a more complete view of the impact of KKE/D domains on nucleolar organization, we took advantage of a recent publication by Tartakoff et al. in which nucleolar proteins were classified into three main categories according to their colocalization with snoRNPs[45–47]. The first group includes proteins that constitutively localize with snoRNPs such as Nsr1. The second group consists of proteins that colocalize with snoRNPs only during exponential growth but not following stress. This group includes early factors such as components of the UTP complexes (Utp22, Utp25), helicases or other AMFs (Rrp5). The third group is composed of late factors such as Enp1, Fpr3 or Rrp1 that never colocalize with snoRNPs in both exponential growth or during stress[45–47]. Interestingly, we showed that the nucleolar accumulation of Nsr1 and Rrp5 was affected by KKE/D domain truncation during exponential growth, as shown by the increased nucleoplasmic signals. In contrast, the localization of the other tested early or late factors remained unchanged (Supplementary Fig. 8A, B). Upon rapamycin treatment, compaction of the Rrp5 signal was not affected by the absence of the snoRNP KKE/D domains, nor was that of all the other tested factors except Nsr1. Compaction of the Nsr1 protein, which was reported to colocalize with snoRNPs in this condition, was strongly affected, supporting again that KKE/D domains are essential for formation of this subnucleolar domain and the sequestration of a specific pool of abundant nucleolar proteins including snoRNPs, RNAPI and other ribosome biogenesis factors such as Nsr1 (Supplementary Fig. 8A, B).

## KKE/D domains mediate efficient snoRNP condensation in a manner dependent on rRNA synthesis

Our data support the hypothesis that under conditions of active rRNA production, the KKE/D domains promote the recruitment of snoRNPs and other AMFs onto the nascent rRNAs, while cooperative self-interactions of KKE/D domains in latent snoRNPs promote formation of a nucleolar condensate when ribosome biogenesis declines. To finally confirm this dependency on rRNA levels and rule out a potential direct effect of TORC1 inactivation on nucleolar organization[52], we reduced rRNA synthesis by directly inhibiting RNAPI using a mutant strain expressing a temperature-sensitive version of the transcription

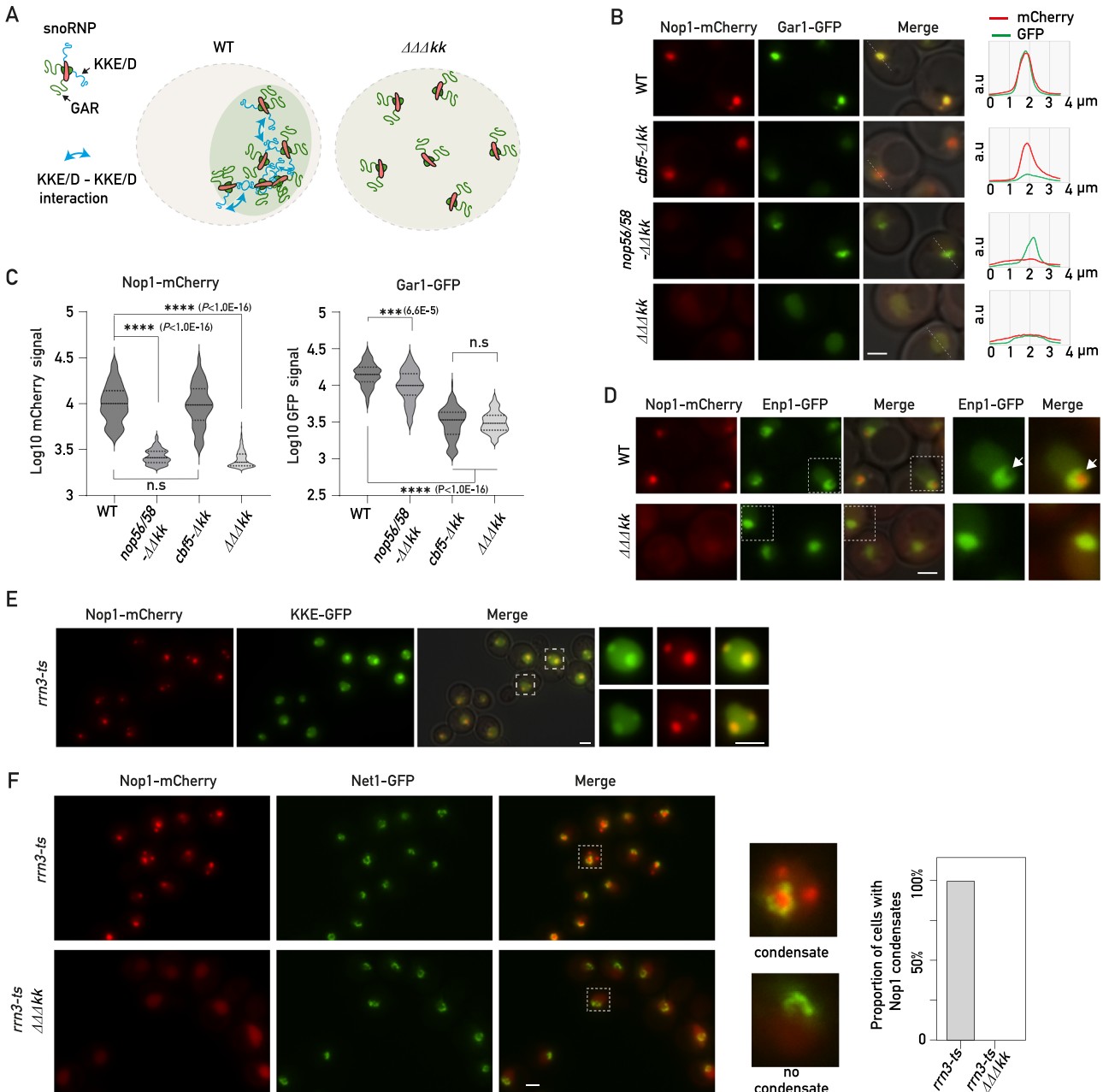

**Fig. 7 | KKE/D domains are essential for nucleolar compaction and sequestration of associated factors in a specific subnucleolar area following TORC1 inactivation. A** Schematic representation of the proposed role of KKE/D domains as dimeric ligands between latent snoRNPs contributing to increase the local concentration of GAR domain-containing proteins in wild-type cells but not in the absence of the KKE/D domains of snoRNPs (*ΔΔΔkk*). **B** Rapamycin-treated wild-type (WT), *cbf5-Δkk, nop56/58-ΔΔkk* or *cbf5-Δkk nop56/58-ΔΔkk (ΔΔΔkk)* cells bearing two plasmids allowing expression of Nop1-mCherry and Gar1-GFP were analyzed by fluorescence microscopy. Average plots (right panel) show the mCherry and GFP signal profiles in the nucleolus along the indicated dotted lines. Scale bar = 2 μm. **C** Quantification of maximum nuclear Nop1-mCherry (left panel: WT (n = 135), *cbf5-Δkk* (n = 64), *nop56/58-ΔΔkk* (n = 142) or *cbf5-Δkk nop56/58-ΔΔkk (ΔΔΔkk*; n = 211)) or Gar1-GFP (right panel: WT (n = 89), *cbf5-Δkk* (n = 68), *nop56/58-ΔΔkk* (n = 100) or *cbf5-Δkk nop56/58-ΔΔkk (ΔΔΔkk*; n = 144)) signals (Log10) in rapamycin-treated cells inspected in (**B**). p values were calculated with unpaired two-tailed Welch's *t* test. n = number of cells pooled from 3 biologically independent replicates.

**D** Rapamycin-treated wild-type (WT) or *ΔΔΔkk* cells expressing Nop1-mCherry and Enp1-GFP were analyzed by fluorescence microscopy. Zooms of the indicated dotted square areas are shown on the right. Arrows show the specific region where Nop1 accumulates in wild-type cells in a representative nucleus. Scale bar = 2 μm. **E** Fluorescence microscopy analyses of *rrn3-ts* cells expressing Nop1-mCherry and the KKE/D-GFP construct (KKE-GFP) grown exponentially at 25 °C and transferred for 1 h to 37 °C. Zooms of the GFP, mCherry and merged signals in the indicated dotted squares are shown on the right. Scale bars = 2 μm. **F** Fluorescence microscopy analyses of *rrn3-ts* or *rrn3-ts ΔΔΔkk* cells expressing Nop1-mCherry and Net1-GFP grown exponentially at 25 °C and transferred for 1 h to 37 °C. Zooms of the merged GFP and mCherry signals within the indicated dotted squares are shown on the right as examples of cells with or without Nop1-mCherry condensates. Scale bar = 2 μm. The right panel shows a proportional graph based on the number of manually counted cells (n = 109 from 3 biologically independent replicates) with or without Nop1-mCherry condensates. Source data are provided as a Source data file.

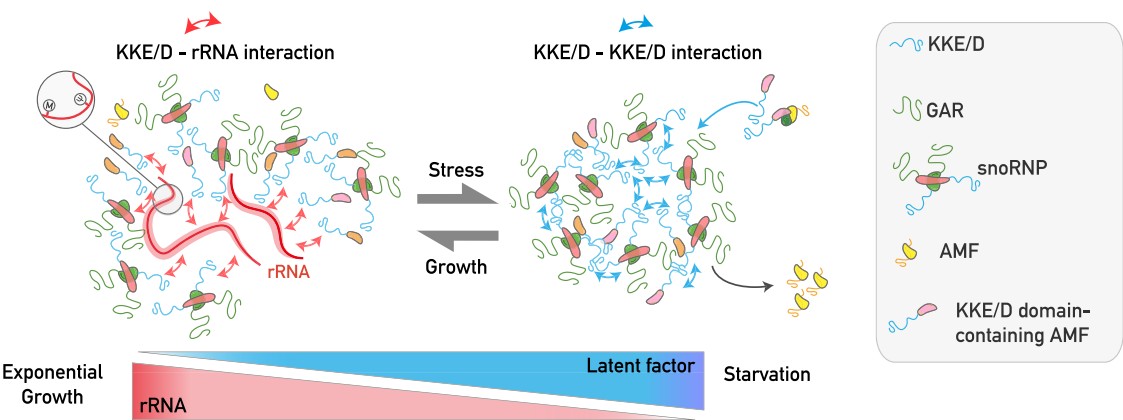

**Fig. 8 | Schematic representation of the proposed model on the dual role of KKE/D domains depending on growth conditions.** During exponential growth, the KKE/D domain is targeted to the close vicinity of actively transcribed rDNA genes through electrostatic interaction with nascent pre-rRNAs, without engaging in cooperative interactions with the other abundant KKE/D domains that are also associated with nascent pre-rRNAs. This specific targeting property is essential for proper activity of snoRNPs and RNA helicases in rRNA modification. In stress conditions, such as RNAPI inhibition or limited nutrient availability, during which the pre-rRNA synthesis rate declines, the proportion of latent KKE/D domain-containing AMFs increases and self-interactions among their KKE/D domains promote the cooperative concentration of these AMFs and their partners in a specific subnucleolar structure. Other early nucleolar AMFs not equipped with a KKE/D domain are excluded from these condensates. In this process, we propose that the KKE/D domains of latent snoRNPs are particularly important to increase the local concentration of GAR domain-containing proteins, allowing formation of subnucleolar condensates and subsequent sequestration of AMFs involved in the earliest stages of ribosome biogenesis (RNAPI, RNA helicases, snoRNPs).

initiation factor Rrn3. In response to RNAPI inhibition, both in yeast and mammalian cells, the snoRNP signal undergoes a global remodeling and fragmentation into one or several spherical condensates in the nucleus (Fig. 7E and Supplementary Fig. 8D)[46,47,53,54]. As expected, the KKE/D-GFP fusion protein was also sequestered in these subnucleolar structures together with Nop1 following RNAPI inhibition (Fig. 7E); however, these subnucleolar condensates completely disappeared in the absence of all KKE/D domains of snoRNPs (Fig. 7F and Supplementary Fig. 8D). Altogether, our data revealed a molecular mechanism by which KKE/D domains either mediate efficient snoRNP activity or regulate the multilayered nucleolar organization depending on rRNA synthesis and the fluctuating pool of latent snoRNPs.

## Discussion

In the last few years, the ability of proteins containing nucleolar IDRs to form biomolecular condensates through LLPS attracted a lot of attention, but the molecular role of IDRs in the cellular context of ribosome biogenesis remains poorly understood. Based on a phenomenological approach to understand the function IDRs in a physiologically relevant context, we revealed a molecular mechanism in which the specific lysine-rich IDR of snoRNPs fulfils a dual, growth-dependent function in early ribosome biogenesis events and the establishment of nucleolar morphology during stress (Fig. 8, see legend for description).

We showed that intrinsic features of KKE/D domains facilitate the co-transcriptional recruitment of snoRNPs and other AMFs in the vicinity of RNAPI. This specific targeting property is essential for proper accumulation of snoRNPs and RNA helicases in the DFC of nucleoli and for their activity in rRNA modification. KKE/D domains most likely provide snoRNPs with the best opportunities in space and time to hybridize with their target sequences and catalyze the modifications before the nascent pre-rRNA transcripts get folded and packaged with AMFs and RPs. The presence of RNA helicases equipped with KKE/D domains in this nucleolar environment likely increases the efficiency of snoRNP dissociation from the pre-rRNA, which is also an important parameter for the efficiency of nucleotide modifications[14]. Intriguingly, we observed that some snoRNP-dependent modifications

at certain positions were drastically reduced, whereas other modification sites were literally unaffected. We propose that depending on whether these target regions are rapidly packaged during the rRNA maturation process or remain longer flexible and accessible within pre-ribosomes, the probability for snoRNPs to hybridize to their target sequences may be different, which may account for the site-specific differences observed in the absence of the snoRNP's KKE/D domains. An alternative hypothesis could be that deletion of the KKE/D domains of snoRNPs affects all nucleotide modifications to the same extent, but that the least functional ribosomes lacking important modifications are rapidly degraded, resulting in a molecular selection of mature ribosomes bearing a specific subset of modifications. Further experiments to measure the modification levels of purified pre-rRNAs associated with RNAPI will be required to challenge these two possibilities.

In stress conditions during which the pre-rRNA synthesis rate declines, the accumulation of latent snoRNPs and self-interactions of their KKE/D domains promote the cooperative concentration of these snoRNPs. This promotes the condensation of GAR domain-containing proteins and the nucleolar sequestration of a specific set of the earliest ribosome biogenesis factors (snoRNPs, RNAPI, RNA helicases and accessory factors) in a specific subnucleolar structure. Other early nucleolar AMFs not equipped with a KKE/D domain are depleted from these condensates. Formation of the subnucleolar condensates gathering the KKE/D domain-containing proteins may allow to sequester early-acting factors in the nucleolar area to avoid their dispersion into the nucleoplasm and the establishment of deleterious non-specific interactions that may interfere with other nuclear processes. In addition, this sequestration may allow to maintain the toolbox of early-acting AMFs (RNA helicases and cofactors thereof, RNAPI and snoRNPs) in the close vicinity of rDNA to keep them poised for association with the nascent pre-rRNA transcript when stress conditions are relieved, thereby facilitating rapid resumption of the earliest steps of ribosome biogenesis. Lastly, considering that all AMFs continuously cycle between latent and active states even in exponential growth phase, this sequestration may promote efficient recycling of active AMFs by avoiding their dispersion following release.

One of the central debates in the field of nucleolar organization and LLPS is to conciliate stable phase-separated compartments with the intense flux of ribosome biogenesis across these phases. Our experiments were not designed to characterize LLPS behavior, but they nevertheless strongly highlight the necessity to differentiate between latent and active states of AMFs to understand the LLPS properties of IDRs. This latter point has recently been discussed by Tartakoff et al.[20]. Importantly, several studies pointed out that although many proteins convincingly form immiscible droplets in one and two-component systems in vitro or after overexpression in vivo, it is less clear to what extent this phenomenon could be physiologically relevant in the crowded cellular context, where the concentration threshold for phase separation determined in vitro is unlikely to be reached[20,23]. An elegant computational study recently proposed that protein condensation by phase-separation in vivo requires specific ligand proteins allowing to locally concentrate IDRs prone to phase separation[50]. In this model, phase separation is particularly favored by the presence of divalent ligands that interact with spacer sites of proteins promoting LLPS. KKE/D domains of snoRNPs fit perfectly with this definition of divalent ligands since KKE/D domains in both H/ACA and C/D snoRNPs are present in two copies, allowing to promote interaction between multiple snoRNP-associated GAR domains, which have been shown to be highly prone to condensation through LLPS mechanisms[22]. Consistent with a role of KKE/D domains in stimulating GAR domain condensation, we have shown that the absence of the KKE/D domains of snoRNPs abolishes condensation of proteins containing GAR domains upon rapamycin treatment. Considering that both lysine-rich extensions and GAR domains are absent from snoRNPs in *archaea*, it is tempting to speculate that these two IDRs may have coevolved to integrate several functional activities essential for the multilayered organization of the nucleolus in physiological and stress conditions. Interestingly, a recent work published during the revision of our study also highlighted the conservation of the lysine-rich IDRs of snoRNPs and RNAPI in yeast and human cells[55,56]. Finally, considering the evolutionary conservation of the molecular features of large lysine-rich IDRs present in eukaryotic snoRNPs, we propose that the dual, growth-dependent role of KKE/D domains might constitute a conserved regulatory system that explains how eukaryotic cells continuously adjust nucleolar morphology and RNAPI activity depending on the fluctuating pool of latent snoRNPs. Given the central role of the nucleolus in ribosome biogenesis, stress response, cell fate decision and disease progression, it will be of great interest to explore this model in human cells and to further investigate the role of other conserved eukaryotic nucleolar IDRs under physiological and stress conditions.

## Methods

### Strains, media and growth conditions

The relevant genotypes of the *S. cerevisiae* strains used in this study are listed in Supplementary Data 6. All strains were derivatives of strain W303. For Y2H analyses, the reporter strain PJ69-4A was used[57]. Plasmids and oligonucleotides used in this study are listed in Supplementary Data 7 and 8, respectively.

Yeast cells were grown at 30 °C in YPD (1% yeast extract, 2% peptone, 2% glucose) medium or in YPG (1% yeast extract, 2% peptone, 2% galactose) or in in SC medium (0.67% nitrogen base without amino acids (BD), 2% dextrose) supplemented with amino acid mixtures. TORC1 inactivation was induced by treating cells grown overnight in SC supplemented with amino acid mixtures for 4 h with rapamycin (Sigma, R8781, 1 mg/ml of stock solution in 90% ethanol, 10% Tween-20) at a final concentration of 200 nM. In case of post-diauxic shift, cells were grown for 2 days at 30 °C in YPD. Routine manipulations (cell growth, transformations, DNA preparations) were carried out using standard procedures. Strains were generated by genomic integration of tagging or disruption cassettes as previously described[58].

### Serial dilution growth assays

Yeast cells were spotted in 10-fold serial dilutions on YPAD (YPD supplemented with 60 μg/ml adenine sulfate) plates, which were incubated at 30 °C or 37 °C for 27 h, 32 h, 48 h, 55 h or 70 h. Solid growth media contained 2% agar. When indicated, BMH-21 (Sigma, 509911) was added to the plates at a final concentration of 15 μM.

### Western blotting

Protein samples were separated on 10% SDS-acrylamide:bisacrylamide (29:1) gels and transferred to nitrocellulose membranes (BioRad) using a Trans-Blot Turbo apparatus (BioRad). Membranes were saturated for 1 h with PBST buffer (137 mM NaCl, 2.7 mM KCl, 10 mM Na$_2$HPO$_4$, 2 mM KH$_2$PO$_4$, 0.1% Tween-20) containing 5% (w/v) powder milk. Following incubation for 2 h with the same buffer containing the primary antibodies, membranes were rinsed three times for 5 min with PBST buffer, incubated for 1 h with the secondary antibodies diluted in PBST containing 5% (w/v) powder milk and finally washed three times for 10 min with PBST buffer. Luminescent signals were generated using the Clarity Western ECL Substrate (Bio-Rad), captured using a ChemiDoc Touch Imaging System (Bio-Rad). HA-tagged proteins were detected using HRP-conjugated mouse monoclonal anti-HA antibodies (Roche Diagnostics, Cat. #12013819001, 1:1000 dilution); GFP-tagged proteins were detected using polyclonal anti-GFP antibodies (1:1000 dilution) generated by custom antibody production services and kindly provided by Marlène Faubladier and Pierre-Emmanuel Gleizes. RNAPI subunits were detected using rabbit polyclonal antibodies detecting all subunits, kindly provided by Michel Riva. Secondary antibodies were purchased from Promega (HRP-conjugated anti-mouse antibodies, Cat. # W402B, 1:10,000 dilution; HRP-conjugated anti-rabbit antibodies, Cat. # W401B, 1:10,000 dilution). Pgk1 was detected using a monoclonal anti-Pgk1 antibody (Invitrogen, Cat. # 2C5D8, 1:5000 dilution).

### Northern blotting

Extraction of yeast total RNA: dry cell pellets were resuspended with 0.5 ml water-saturated phenol and 0.5 ml guanidine thiocyanate (GTC) mix (50 mM Tris-HCl, pH 8.0, 10 mM EDTA, pH 8.0, 4 M guanidine thiocyanate, 2% N-Lauroylsarcosine, 143 mM β-Mercaptoethanol). Cells were broken by vigorous vortexing three times for 2 min at 4 °C in the presence of Zirconium beads. The resulting samples were mixed with 7.5 ml water-saturated phenol and 7.5 ml GTC mix and incubated for 5 min at 65 °C. After addition of 7.5 ml chloroform and 4 ml sodium acetate buffer (10 mM Tris-HCl, pH 8.0, 1 mM EDTA, pH 8.0, 100 mM sodium acetate), samples were mixed vigorously and centrifuged at 3220 × g for 5 min at 4 °C. Aqueous phases were recovered and RNAs were re-extracted three additional times with water-saturated phenol:chloroform (1:1). RNAs were then concentrated by ethanol precipitation and ultimately resuspended in ultrapure H$_2$O. In all northern blotting experiments (see below), equal amounts of these total RNAs (4 μg) were analyzed.

For Northern blotting analyses of high-molecular-mass RNA species, RNAs were separated as described in "Molecular Cloning", Sambrook and Russell, CSHL Press ("Separation of RNA According to Size: Electrophoresis of Glyoxylated RNA through Agarose Gels"). RNAs were then transferred to Hybond N + membranes (GE Healthcare) by capillary using 5× SSC as a transfer buffer. Low-molecular-mass RNA species were separated by electrophoresis through 6% acrylamide:bisacrylamide (19:1), 8 M urea gels using 1× TBE as a running buffer. RNAs were then transferred to Hybond N membranes (GE Healthcare) by electro-transfer in 0.5× TBE buffer, 20 V, 4 °C, overnight. In all cases, membranes were hybridized with $^{32}$P-labeled oligonucleotide probes using Rapid-hyb buffer (GE Healthcare). Radioactive membranes were exposed to Phosphorimaging screens and revealed using Typhoon TRIO or Typhoon 9400 Variable Mode Imagers (GE Healthcare) driven by Typhoon Scanner Control software (Version 5.0). Sequences of the

oligonucleotides used as probes in this study are described in the Supplementary Data 8. Quantifications of the signals were performed using PhosphorImager data and Multi Gauge software (Version 3.0, FUJIFILM). Statistically significant differences were determined using one-tailed Student's $t$ test.

## Electron microscopy

For morphological analysis of nucleoli, budding yeast cells were cryofixed by high pressure freezing (EMPACT; Leica) with liquid nitrogen and cryosubstituted with 0.1% uranyl acetate in anhydrous acetone, at −90 °C for 72 h. Cells were then embedded in a Lowicryl resin (HM-20) polymerized at −50 °C with UV irradiation. Ultrathin sections of 80–100 nm were mounted on 300-mesh nickel grids and contrasted with UranyLess and lead citrate. Grids were examined with a transmission electron microscope (Jeol JEM-1400, JEOL, Inc.) at 80 kV. Images were acquired using a digital camera (Gatan Orius, Gatan, Inc.).

## Correlative light electron microscopy (CLEM)

For CLEM, the procedure for sample cryofixation and cryosubstitution was similar to the one used for studying nucleolar morphology[39]. Next, sections of 150–180 nm were mounted on copper H6 Maxtaform Finder grids coated with Formwar and carbone. The grids were mounted between slide and coverslip in a Citifluor antifadent solution. Acquisition of the fluorescent signal was performed with a Nikon TI-E/B inverted microscope featuring an EMCCD camera (Ixon Ultra DU897-ANDOR) and a HG intensilight illumination. Images were acquired using Nikon CFI Apo TIRF 100X (NA = 1.49) objective and Semrock filters sets for GFP (Ex: 482BP35; DM: 506; Em: 536BP40). Light transmitted images of the microscope fields of interest were also acquired in order to locate the cells of interest using the alphanumerical code of the Finder Grids. The grids were subsequently detached, thoroughly rinsed in large volumes of ultrapure-Q water and air-dried. The sections were next contrasted with UranyLess and lead citrate. Grids were examined with a Jeol 1200X electron microscope (Jeol JEM-1400, JEOL, Inc. at 80 kV. Thanks to the alphanumerical code present in Finder grids, the regions of interest where fluorescent signals were detected were spotted. Images were acquired using a digital camera (Gatan Orius, Gatan, Inc.).

## Fluorescence microscopy, quantifications, and statistical analyses

Cells were grown overnight at 30 °C in SC medium (0.67% nitrogen base without amino acids (BD), 2% dextrose) supplemented with amino acid mixtures. Cells were diluted and were harvested when $OD_{600}$ reached 0.4. Yeast culture was treated as indicated in the Figure legends and 2 µl aliquots of cell suspensions were spotted on a microscope slide containing a slab of 2% agarose + SC medium complemented with amino acids and glucose. An inverted wide field microscope Nikon Ti Eclipse equipped with an EM-CCD camera and a thermostatic chamber was used for acquisition. Images were captured and processed using ImageJ.

The contrast was systematically adjusted in a similar way between the images shown in the same panel, except in Fig. 6D and Supplementary Fig. 6E, as indicated in the Figure legend. Each microscopic observation was reproduced several times ($n > 3$) on different days.

To quantify the maximum nuclear signal in the nucleolus, several hundred cells were cropped using a similar threshold between the compared growth conditions or mutant cells in order to define the nucleolus or the nucleus and the maximum signal for each channel in a same region of interest was calculated using ImageJ. The data were next plotted in a violin plot. $p$ values were calculated with unpaired two-tailed Welch's $t$ test and $n$ indicates the number of cells, pooled from images from at least three independent biological replicates. ($p$ value 0.05 < *; 0.01 < **; 0.001 < *** and 0.0001 < ****, ns not significant).

In Supplementary Fig. 7F, we used the Net1-mKate signal to localize and measure the GFP nucleolar signal, and the nuclear background of GFP was used to identify the nucleoplasm. The ratios of GFP intensity between these two regions of interest were calculated using ImageJ. The data were next plotted in a violin plot. $p$ values were calculated as described above.

## RiboMeth-Seq and HydraPsi-Seq experiments

Total RNA from $S. cerevisiae$ cells was isolated as described above. RNA concentration was measured by Nanodrop One and RNA quality was checked by automated electrophoresis on 4150 TapeStation system (Agilent technologies, USA). RNA (100–200 ng) was subjected to either RiboMethSeq or HydraPsiSeq protocols.

For RMS treatment, RNA was fragmented by alkaline hydrolysis in 50 mM bicarbonate buffer pH 9.2 for 16 min at 96 °C. The reaction was stopped by ethanol precipitation using 0.3 M Na-OAc, pH 5.2 and glycoblue in liquid nitrogen. After centrifugation, the pellet was washed with 80% ethanol and resuspended in nuclease-free water. RNA fragments were first dephosphorylated using 5 U of Antarctic Phosphatase (New England Biolabs, UK) for 30 min at 37 °C followed by 5 min at 70 °C in order to inactivate the phosphatase. RNA fragments were then phosphorylated at the 5′-end using 10 U of T4 PNK and 1 mM ATP for 1 h at 37 °C. End-repaired RNA fragments were purified using RNeasy MinElute Cleanup kit (QIAGEN, Germany). To bind small RNA fragments to the RNeasy MinElute columns, 675 µl of 96% ethanol were used for RNA binding. Elution was performed in 10 µl of nuclease-free water.

For HPS, RNA was subjected to hydrazine treatment (50% final concentration) for 45 min on ice. The reaction was stopped by ethanol precipitation using 0.3 M Na-OAc, pH 5.2 and glycoblue at −80 °C for 2 h. After centrifugation, the pellet was washed twice with 80% ethanol and resuspended in 1 M aniline, pH 4.5. The reaction was incubated in the dark for 15 min at 60 °C and precipitated using 0.3 M Na-OAc, pH 5.2 and glycoblue at −80 °C overnight. After centrifugation, the pellet was washed twice with 80% ethanol and RNA was dephosphorylated at the 3′-end using 10 U of T4 PNK in 100 mM Tris–HCl pH 6.5, 100 mM Mg-OAc and 5 mM β-mercaptoethanol and incubated for 6 h at 37 °C. T4 PNK was inactivated by incubation for 20 min at 65 °C. RNA was purified using RNeasy MinElute Cleanup kit (QIAGEN, Germany) as previously described for RMS experiments. Elution was performed in 10 µl of nuclease-free water. For both RMS and HPS experiments, RNA fragments were converted to a library using NEBNext® Small RNA Library kit (New England Biolabs, UK) using the manufacturer's instructions. DNA library quality was assessed using a High Sensitivity DNA chip on a 4150 Tapestation system (Agilent technologies, USA). Library quantification was done using a fluorometer (Qubit 3.0 fluorometer, Invitrogen, USA). Libraries were multiplexed and subjected to high-throughput sequencing using a NextSeq 2000 instrument with 50 bp single read runs. Libraries were loaded onto the flow cell at 650 pM final concentration. Analysis and quantification of 2′-O-methylations and pseudouridine residues was performed essentially according to previously published protocols[59–61]. In brief, raw reads were processed to remove eventual sequences of Illumina adapter and, for RiboMethSeq protocol, short reads <40 nt were selected to capture exact positions of both 5′- and 3′-ends. Alignment was performed on the mature sequences of yeast rRNA. After *.bam conversion to *.bed format, both 5′- and 3′-ends (or only 5′-end for HydraPsiSeq) were counted using custom awk script. Combined 5′/3′-end count was used for calculation of RiboMethSeq scores and MethScore (also known as ScoreC2 for +/− 2 nt window) was used for quantification of the methylation level at all known Nm positions in yeast $S. cerevisiae$ rRNA. HydraPsiSeq data were treated in a similar way, but only 5′-end count was used to establish raw U cleavage profile. Normalization to random cleavages observed for A, C and G nucleotides in 10 nt window was used to create NormUcount profile. Non-U nucleotides were dropped

and PsiScore (conceptually identical to MethScore used in Ribo-MethSeq) was used for quantification of the pseudouridylation level.

## Immunoprecipiation of native pre-ribosomal particles

Cell pellets corresponding to 500 ml cultures at OD$_{600}$ ~ 0.6 were resuspended with approximately one volume of ice-cold A200-KCl buffer (20 mM Tris-HCl, pH 8.0, 5 mM magnesium acetate, 200 mM KCl, 0.2% Triton X-100) supplemented with 1 mM DTT, 1× Complete EDTA-free protease inhibitor cocktail (Roche), 0.1 U/μl RNasin (Promega). About 400 μl of ice-cold zirconia beads were added to 800 μl aliquots of the resuspended cells, which were broken by vigorous shaking, two times 30 sec separated by 1 min incubation on ice using a Precellys 24 apparatus (Bertin). Extracts were clarified through two successive centrifugations at 16,000 × g and 4 °C for 5 min and quantified by measuring absorbance at 260 nm. Equal amounts of soluble extracts were incubated for 2 h at 4 °C with polyclonal anti-GFP antibodies in a total volume of 1 ml (adjusted with A200-KCl buffer supplemented with 1 mM DTT, 1× Complete EDTA-free protease inhibitor cocktail, 0.1 U/μl RNasin) on a rocking table. The immune complexes were recovered using 20 μl of magnetic Bio-Adembeads conjugated to proteins A/G (04631, Ademtech) that were added to each sample and incubated for 1 h at 4 °C on a wheel. Beads were washed seven times with 1 ml of ice-cold A200-KCl buffer supplemented with 1 mM DTT. RNAs were extracted from bead pellets as follows: 160 μl of 4 M guanidium isothiocyanate solution, 4 μl of glycogen (Roche), 80 μl of [10 mM Tris-HCl, pH 8.0, 1 mM EDTA, pH 8.0, 100 mM sodium acetate], 120 μl of phenol and 120 μl of chloroform were added. Tubes were shaken vigorously, incubated 5 min at 65 °C, and centrifuged 5 min at 16,000 × g (4 °C). Aqueous phases (240 μl) were mixed vigorously with 120 μl of phenol and 120 μl of chloroform, centrifuged 5 min at 16,000 × g (4 °C) and the resulting aqueous phases were ethanol precipitated.

## Chromatin immunoprecipitation (ChIP)

Exponentially growing cells (50 ml cultures) were crosslinked for 15 min at 30 °C with 1% formaldehyde (Sigma) and the crosslinking reaction was quenched with 125 mM glycine for 5 min at 30 °C. Cells were harvested by centrifugation, cell pellets were washed twice with 20 ml ice-cold 1× PBS [137 mM NaCl, 2.7 mM KCl, 10 mM Na$_2$HPO$_4$, 2 mM KH$_2$PO$_4$], resuspended with 1 ml ice-cold 1× PBS, transferred into screw-cap microtubes, centrifuged again and frozen. Cells were resuspended with 400 μl lysis buffer (50 mM HEPES-KOH pH 7.5, 500 mM NaCl, 1 mM EDTA, 1% Triton X-100, 0.1% sodium-deoxycholate, 0.1% SDS) supplemented with 1× complete EDTA-free protease inhibitor cocktail (Roche). About 500 μl of ice-cold glass beads were added and cells were broken using a Precellys 24 apparatus (Bertin), 2 times 20 sec at 5500 rpm with 1 min pause in between. Chromatin was recovered by centrifugation at 1150 × g, 5 min, 4 °C, resuspended with 1400 μl lysis buffer supplemented with 1× complete EDTA-free protease inhibitor cocktail (Roche), transferred into 15 ml Falcon tubes and sheared by sonication using a Branson Sonifier, 5 times 24 pulses (with incubations on ice in between), output 4 at 50% duty cycle. Sonicated chromatin was transferred into 1.5 ml microtubes, centrifuged 5 min at 15,871 × g and 4 °C and supernatants (soluble chromatin fragments) were transferred into new microtubes. Nucleic acid concentration (absorbance at 260 nm) of the samples was measured using Nanodrop and samples corresponding to about 200 μg of nucleic acids were incubated with 2 μg mouse anti-HA antibodies (16B12, MMS-101R, Covance) or 2 μg of rabbit anti-GFP antibodies (kindly provided by Marlène Faubladier and Pierre-Emmanuel Gleizes) and incubated overnight at 4 °C on a wheel. The immune complexes were recovered using 20 μl of magnetic Bio-Adembeads conjugated to proteins A/G, (04631, Ademtech) that were added to each sample and incubated for 2 h at 4 °C in a Thermomixer Comfort (Eppendorf) with shaking at 1000 rpm. Magnetic beads were washed

twice for 10 min at 4 °C on a rocking table with 1 ml lysis buffer, once with 1 ml sodium-deoxycholate buffer (10 mM Tris-HCl pH 8.0, 1 mM EDTA pH 8.0, 50 mM LiCl, 0.5% NP-40, 0.5% sodium-deoxycholate) and once with 1 ml TE buffer (50 mM Tris-HCl pH 8.0, 10 mM EDTA pH 8.0). To elute the immunoprecipitated material, magnetic beads were resuspended with 50 μl elution buffer (50 mM Tris-HCl pH 8.0, 10 mM EDTA pH 8.0, 1% SDS) and incubated 15 min at 37 °C in a Thermomixer Comfort (Eppendorf) with shaking at 1300 rpm. For protein analyses by western blotting, one fifth of the eluate volumes (10 μl) were mixed with 10 μl of 2× protein loading buffer (100 mM Tris-HCl pH 6.8, 4% SDS, 20% Glycerol, 0.2% Bromophenol blue, 200 mM DTT) and incubated two times for 15 min at 95 °C (Eppendorf Thermomixer Comfort) with a centrifugation pulse in between to avoid substantial evaporation. For DNA analyses by qPCR, the remaining eluate volumes (40 μl) were incubated overnight at 65 °C (hybridization oven) to reverse crosslinks. To degrade proteins, 10 μg of proteinase K (Roche) and 10 μg of glycogen (Roche) were added in a final volume of 100 μl of TE and samples were incubated 2.5 h at 56 °C (hybridization oven). Nucleic acids were then purified by phenol:chloroform:isoamyl alcohol extraction and ethanol precipitation. Final nucleic acid pellets were resuspended with 20 μl ultrapure H$_2$O and RNAs were degraded with 10 μg of RNase A for 1 h at 37 °C.

## qPCR analyses

Ct values for each sample (IP and Input) were determined by median of triplicate qPCR experiments. The Ct values were then used to calculate the fold difference between IP and normalized input. ΔCt [normalized ChIP] = (Ct [ChIP] − (Ct [Input] − Log2 (1/500 (dilution factor)). Here, the dilution factor of the Input corresponds to 10% of the chromatin fraction used for the IP, and the input was next diluted 50 times prior to qPCR, resulting in a final dilution factor of 1/500 relative to the IP. Finally, the IP/Input ratio values were calculated as follows: ratio IP/Input = 1/2^ ΔCt [normalized ChIP].

## In vitro coacervate formation and analysis

The [KKE/D]x9 peptide (N-KKEKKEKKDKKDKKEKKEKKDKKDKKEK-C) conjugated or not to Alexa 488 (AF 488 Maleimide) at the N-terminus was synthesized by Thermofisher Peptide Custom Synthesis Service and solubilized at 10 mg/ml in 50 mM Tris-HCl pH 8. Mixtures of fluorescent and unlabeled [KKE/D]x9 peptides were prepared at a ratio of 1:20 in 50 mM Tris-HCl pH 8, 7.5% PEG, at a final concentration of 150 μM in the absence or presence of 50 μg/ml poly-uridylic acid potassium salt (poly-U RNA, MW = 600–1000 kDa, ~ 2000–3300 residues from Sigma-Aldrich) or 50 μg/ml of baker's yeast total RNAs. If stated, NaCl was added at a final concentration of 300 mM. After 15 min incubation at room temperature, 5 μl of suspensions were spotted onto a microscope slide and imaged using an inverted wide field microscope Nikon Ti Eclipse equipped with an EM-CCD camera in a thermostatic chamber. Images were captured and processed using ImageJ. To test the stability of the KKE/D domain condensates, KKE/D coacervates were first preformed (by incubating 15 min at room temperature a mix of 150 μM KKE/D peptides in 50 mM Tris-HCl pH 8, 7.5% PEG, 50 μg/ml (poly-U RNA) or 50 μg/ml of baker's yeast total RNAs) followed by the addition of NaCl in a Tris-buffered solution (50 mM Tris-HCl pH 8) at a final concentration of 300 mM. 5 μl of suspensions were spotted onto a microscope slide 60 minutes after NaCl addition and imaged as previously described.

## Identification of protein neighborhoods by TurboID-based proximity labeling

Plasmids expressing the N-terminally TurboID-tagged Cbf5(392-483) bait protein and the SV40 NLS-TurboID-GFP control protein under the control of the copper-inducible *CUP1* promoter were transformed into the wild-type strain YDK11-5A. Transformed cells were grown at 30 °C in 100 ml SC-Leu medium, containing copper-free yeast nitrogen base

(FORMEDIUM), to an $OD_{600}$ of around 0.6–0.8. Then, copper sulfate, to induce expression from the *CUP1* promoter, and freshly prepared biotin (Sigma-Aldrich) were added to a final concentration of 500 µM, and cells were grown for an additional hour, typically reaching a final $OD_{600}$ between 0.8 and 1.0, and harvested by centrifugation at $3220 \times g$ for 5 min at 4 °C. Then, cells were washed with 50 ml ice-cold $H_2O$, resuspended in 1 ml ice-cold lysis buffer (LB: 50 mM Tris-HCl pH 7.5, 150 mM NaCl, 1.5 mM $MgCl_2$, 0.1% SDS, and 1% Triton X-100) containing 1 mM PMSF, transferred to 2 ml safe-lock tubes, pelleted by centrifugation, frozen in liquid nitrogen, and stored at −80 °C.

Cells were resuspended in 400 µl lysis buffer containing 0.5% sodium deoxycholate and 1 mM PMSF (LB-P/D), and cellular extracts were prepared by glass bead lysis with a Precellys 24 homogenizer (Bertin Technologies) set at 5000 rpm using a $3 \times 30$ s lysis cycle with 30 s breaks in between at 4 °C. Lysates were transferred to 1.5 ml tubes. For complete extract recovery, 200 µl LB-P/D were added to the glass beads and, after brief vortexing, combined with the already transferred lysate. Upon clarification by centrifugation for 10 min at $17,115 \times g$ at 4 °C, cell lysates were transferred to a new 1.5 ml tube, and the volume was completed to around 800 µl by the addition of around 200 µl LB-P/D. To reduce non-specific binding, 30 µl of Pierce Streptavidin Magnetic Beads (Thermo Scientific), corresponding to 0.3 mg of settled beads, were transferred to a 1.5 ml safe-lock tube, blocked by incubation with 900 µl LB containing 3% BSA for 1 h at RT. For affinity purification of biotinylated proteins, 20 $A_{260}$ units of cell lysate in an adjusted volume of 800 µl LB-P/D was added to the blocked streptavidin beads, and binding was carried out for 2 h at RT on a rotating wheel. Beads were then washed once for 5 min on a rotating wheel with 900 µl of wash buffer 1 (50 mM Tris-HCl pH 7.5, 2% SDS), three times with 900 µl LB, and finally three times with 900 µl of wash buffer 2 (50 mM Tris-HCl pH 7.5). To elute, reduce, and alkylate the bound proteins, the beads were resuspended in 50 µl of resuspension buffer (100 mM Tris-HCl pH 8, 8 M urea, 10 mM biotin, 1 mM DTT) and incubated for 10 min at RT; then, 1 µl 550 mM iodoacetamide was added, and the mixture was incubated in a thermoshaker, set to 1200 rpm, for 3 h at 25 °C in the dark. For in-solution digestion, the urea concentration was first reduced to 1 M by adding 50 µl $H_2O$ and 300 µl of a 50 mM Tris-HCl pH 8 solution; then, 10 µl of a 50 mM Tris-HCl pH 8 solution containing 2 µg of Sequencing Grade Modified Trypsin (Promega) was added. Digestion of proteins was carried out overnight at RT on a rotating wheel. To stop the digestion, 5 µl of a 50% trifluoroacetic acid (TFA) solution was added. For desalting and peptide purification, the samples were applied to C18 StageTips[62], equilibrated with 50 µl of buffer B (80% acetonitrile, 0.3% TFA) and washed twice with 50 µl of buffer A (0.5% acetic acid). StageTips were washed once with 100 µl of buffer A, and the peptides were eluted with 50 µl of buffer B. The solvents were completely evaporated using a SpeedVac, and the peptides were resuspended in 20 µl of buffer A*/A (30% buffer A* (3% acetonitrile, 0.3% TFA) / 70% buffer A) and stored at −80 °C.

LC-MS/MS measurements were performed on a Q Exactive HF-X (Thermo Scientific) coupled to an EASY-nLC 1000 nanoflow-HPLC (Thermo Scientific). HPLC-column tips (fused silica) with 75 µm inner diameter were self-packed with ReproSil-Pur 120 C18-AQ, 1.9 µm particle size (Dr. Maisch GmbH) to a length of 20 cm. Samples were directly applied onto the column without a pre-column. A gradient of A (0.1% formic acid in $H_2O$) and B (0.1% formic acid in 80% acetonitrile in $H_2O$) with increasing organic proportion was used for peptide separation (loading of sample with 0% B; separation ramp: from 5–30% B within 85 min). The flow rate was 250 nl/min and for sample application 600 nl/min. The mass spectrometer was operated in the data-dependent mode and switched automatically between MS (max. of $3 \times 10^6$ ions) and MS/MS. Each MS scan was followed by a maximum of ten MS/MS scans using normalized collision energy of 28% and a target value of 10,000. Parent ions with a charge state form $z = 1$ and unassigned charge states were excluded for fragmentation. The mass range for MS was $m/z = 370$–1750. The resolution for MS was set to 120,000 and for MS/MS to 30,000. MS parameters were as follows: spray voltage 2.3 kV, no sheath and auxiliary gas flow, ion-transfer tube temperature 250 °C.

The MS raw data files were analyzed with the MaxQuant software package version 1.6.2.10[63] for peak detection, generation of peak lists of mass-error-corrected peptides, and database searches. The UniProt yeast *Saccharomyces cerevisiae* database (version November 2021), additionally including common contaminants, trypsin, TurboID, and GFP, was used as reference. Carbamidomethylcysteine was set as fixed modification and protein amino-terminal acetylation, oxidation of methionine, and biotin were set as variable modifications. Four missed cleavages were allowed, enzyme specificity was Trypsin/P, and the MS/MS tolerance was set to 20 ppm. Peptide lists were further used by MaxQuant to identify and relatively quantify proteins using the following parameters: peptide and protein false discovery rates, based on a forward-reverse database, were set to 0.01, minimum peptide length was set to seven, and minimum number of unique peptides for identification and quantification of proteins was set to one. The "match-between-run" option (0.7 min) was used.

For quantification, missing iBAQ (intensity-based absolute quantification) values in the control purification from cells expressing the SV40 NLS-TurboID-GFP fusion protein were imputed in Perseus[63]. For normalization of intensities in each independent purification, iBAQ values were divided by the median iBAQ value, derived from all nonzero values, of the respective purification. To calculate the enrichment of a given protein compared to its abundance in the control purification, the normalized iBAQ values were log2 transformed and those of the control purification were subtracted from the ones of the TurboID-Cbf5(392-483) bait purification (non-normalized data are available in Supplementary Data 9). For graphical representation ($n = 1$; Source data are provided as a Source Data file.), the normalized iBAQ value (log10 scale) of each protein detected in the TurboID-Cbf5(392-483) bait purification was plotted against its relative abundance (log2-transformed enrichment compared to the control purification).

### Y2H interaction analysis

For Y2H-interaction assays, plasmids expressing bait proteins, fused to the Gal4 DNA-binding domain (Gal4BD), and prey proteins, fused to the Gal4 activation domain (Gal4AD), were co-transformed into reporter strain PJ69-4A. Y2H interactions were documented by spotting representative transformants in 10-fold serial dilution steps onto SC-Trp-Leu (TL), SC-Trp-Leu-His (TLH; *HIS3* reporter), and SC-Trp-Leu-Ade (TLA; *ADE2* reporter) plates, which were incubated for 3 days at 30 °C. Growth on SC-Trp-Leu-His plates is indicative of a weak/moderate interaction, whereas only relatively strong interactions permit growth on SC-Trp-Leu-Ade plates.

### Cloning of the mutKKE and IDR-enp1

The mutKKE IDR amino acid sequence was first designed in silico by mutating charged amino acids (mainly K and E residues). The choice of mutated amino acids has been done in order to (1) remove the charged nature of the KKE/D domain, (2) maintain the flexibility and size of this IDR, and (3) avoid non-physiological large repetitions of similar amino acids that could affect translation and/or aggregation. This sequence was then optimized in terms of codons for yeast and synthesized (Eurofins).

(sequence:GGAGCAGGGGCTAATAATAACATGAAGAACACAAA-CAATAATGGTCAGCAGGCAAAGGAGAACTCATTGATAAAA-GAAGTCTCTACTGAAAAAAAATGGTGTTAAA-GAAGCTCACTCGCAGGGCGCTGCCGCAGAACAACAA-GACGCCGCCGATGCGCAAAATAATCAAGCGCAAGCA-CAAGCCGGCAACTCCGGTAGTGCTCAAAGCTCA-GATGGTCATTCCTCTAATGGTGCAGGAGCTGGAGCTATTTTTAgg) and

amplified with the primer MutKKE_IF_For, MutKKE_IF_Rev (Supplementary Data 8).

Enp1-IDR (amino acids from 28 to 148) was amplified by PCR from W303 background genomic DNA using primers ENP_IF_For, ENP_IF_Rev (Supplementary Data 8). These two IDRs (Enp1 and mutKKE) were cloned by In-SupplementaryFusion® Cloning (Takara) into PacI-digested pFA6a-GFP-kanMX6 (plasmid #39292, addgene). pFA6a-mutKKE-GFP-kanMX6 or pFA6a-IDR-enp1-GFP-kanMX6 were then used to amplify mutKKE or Enp1-IDR fused to GFP and kanMX6, respectively, using INS_mutKKE and CFB5_R1 primers or INS_IDR-Enp1 and CFB5_R1 primers. These PCR fragments were then integrated in place of the KKE/D domain by homologous recombination as previously described[58].

## Statistic and reproducibility

Detailed information for each experiment, including the exact number of samples (*n*) and *p* value, is provided in the figure legends, this section, and the source data file. Sample size (*n*) was determined for each set of experiments to reveal reproducibility and/or statistical significance of the data. Sample sizes for RiboMeth-seq and hydra-psy seq experiments was limited to three or four biological replicates and does not require more due to high reproducibility. Although we assumed a normal distribution of the large (*n* > 40) microscopy quantification data set, formal tests and validation of this assumption were performed. The method of data normality we used to validate the normal distribution of our samples is the relative value of the standard deviation to the mean. The standard deviation was systematically less than half the mean, indicating that the data are considered normal. Live imaging was performed from random locations on the coverslips of three biologically independent replicates to minimize the effect of covariates. Randomization was not performed during biochemical and cell biological experiments. All comparative experiments were performed using cells showing no major cell growth defect and all experiments were performed with appropriate controls and established conditions to minimize the impact of covariates. All data are provided in the source data file.

All box plots (Figs. 1D, 2C–G, 4H and Supplementary Fig. 4C) display medians at their centers and are enclosed by the first and third quartiles. Whiskers extend to 1.5 times the interquartile range on both ends.

All violin plots (Figs. 3C, 6C, G, 7C and Supplementary Figs. 3C, 6A, G, I, 7G, 8B) comprise a density plot, the width of which indicates the frequency and display medians at their centers. The first and third quartiles are marked with a line.

All statistical analyses shown in this study were based on at least three independent replicates. In all figures, significant differences are indicated by stars (*p* value 0.05 < *; 0.01 < **; 0.001 < *** and 0.0001 < ****) and with the precise *p* value on the graph (*p*).

Each microscopic observation, western blot or northern blot was reproduced several times on different days in Figs. 1I (*n* = 3), 1J (*n* = 3), 2B (*n* = 2), 3B (*n* = 3), 3F (top panel (*n* = 4), bottom panels (*n* = 2), 4A (*n* = 5), 4D (*n* = 2), 4E (*n* = 2), 6A (*n* = 4), 6B (*n* = 3), 6D (*n* = 3), 6E (*n* = 3), 6F (*n* = 3), 7B (*n* = 3), 7D (*n* = 3), 7E (*n* = 3), 7F (*n* = 3) and Supplementary Figs. 1F (*n* = 2), 3A (*n* = 3), 3B (*n* = 3), 3F (*n* = 2), 4A (*n* = 3), 6C (*n* = 3), 6D (*n* = 2), 6F (*n* = 3), 6H (*n* = 3), 7D (*n* = 3), 7F (*n* = 3), 8A (*n* = 3), 8D (*n* = 4).

## Published data used in this study

The list of all *S. cerevisiae* IDRs determined in ref. 64 was used to identify IDRs longer than 30 amino acids of assembly and maturation factors. NCPR and FCR values were determined as defined in refs. 65,66; protein abundance was determined according to refs. 65,66. Net charge per residue distribution (Fig. 1G) and phase diagram (Fig. 1B and Supplementary Figs. 1D, 7C) is obtained using CIDER (http://pappulab.wustl.edu/CIDER/analysis/). CIDER Classification of Intrinsically Disordered Ensemble Regions is a web-server

developed by the Pappu Lab for calculating parameters relating to disordered protein sequences. The UniProt Knowledgebase[67] was used to search for annotated Cbf5 orthologs. Yeast GFP database[68] has been used to assign proteins to nuclear or nucleolar localization and[41] has been used to assign proteins that co-purify with RNAPI (Fig. 4).

## Reporting summary

Further information on research design is available in the Nature Portfolio Reporting Summary linked to this article.

## Data availability

The MS proteomics data have been deposited to the ProteomeXchange Consortium via the PRIDE partner repository with the dataset identifier PXD056946. Raw sequencing data analyzed this study have been deposited in the ENA (European Nucleotide Archive) under the accession codes PRJEB67499 (RMS data in wild-type or mutant cells grown on glucose (WT, rpa34-Δkk, nop56/58-ΔΔk, cbf5-Δkk and pxr1-Δkk tma23-Δkk) or galactose (WT Gal and rdn1Δ pGAL::rDNA rpa135Δ)) and PRJEB67500 (HPS data in wild-type or mutant cells grown on glucose (WT, rpa34-Δkk, nop56/58-ΔΔk, cbf5-Δkk and pxr1-Δkk tma23-Δkk) or galactose (WT Gal and rdn1Δ pGAL::rDNA rpa135Δ)). Source data are provided with this paper.

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

## Acknowledgements

This work was supported by ANR-21-CE12-0008-01, Unité de biologie moléculaire, cellulaire et du développement (to B.A. and N.K.M.), ANR-16-CE12-0027-01(CHROMRIB, to A.K.H.), Swiss National Science Foun-dation (SNSF) project grants 31003A_175547 and 310030_204801 (to D.K.), the "Fondation ARC pour la Recherche sur le Cancer" (ARCPJA 20191209547 and ARCPJA22020060002067) and the "Ligue Nationale Contre le Cancer" (3FI14194UPAL). We acknowledge the LITC and METI imaging facilities of the Centre de Biologie Intégrative (CBI), member of the national infrastructure France-BioImaging supported by the French National Research Agency (ANR-10-INBS-04). We are grateful to Sylvain Cantaloube from LITC facility (Light Imaging Toulouse CBI) for technical advice and helpful discussions. Electron microscopy was performed at the METI facility (Multiscale Electron Imaging CBI) with the help of Ste-phanie Balor and Vanessa Soldan. RiboMeth-seq and HydraPsi-seq were performed at the Epitranscriptomics & Sequencing (EpiRNA-Seq) in Nancy. We thank Devanarayanan Siva Sankar for measuring the TurboID samples and performing the data analysis in MaxQuant. We are also grateful to Fabian Erdel, Fernando Muzzopappa, Cheryn Ali and David Depierre for technical advice and helpful discussions. We thank all members of the Henry/Henras team for helpful discussions and support. We are grateful to Christine Maheu for technical support. Our work has also benefited from fruitful technical and scientific contacts with the neighboring CBI groups and with members of BBF.

## Author contributions

O.G., A.K.H., and B.A. conceived the study. C.D., N.K.M., A.K.H., and B.A. performed most experiments and analyzed the results together with O.G., A.K.H., B.A., and D.K. A.M.G. and D.K. designed and constructed the Y2H plasmids and tested interactions. A.M.G. and B.P. performed the TurboID assays and analyzed the results together with D.K. V.M. and Y.M. generated the HPS-seq and RMS-seq data and analyzed the results together with O.G., A.K.H., and B.A. H.H., C.D., and Y.H. performed Northern blotting analyses. I.L.S. performed electronic microscopy analyses. V.G.N. constructed some strains presented in Fig. 8A. B.A. and A.H. wrote the manuscript with contributions from O.G. and D.K.

## Competing interests

The authors declare no competing interests.
