## [Transparent Peer Review file · Nature Communications]

The dual life of disordered lysine-rich domains of snoRNPs in rRNA modification and nucleolar compaction

Corresponding Author: Dr Benjamin Albert

Version 0:

Reviewer comments:

Reviewer #1

(Remarks to the Author)

This study presents a systematic analysis of the downstream effects of perturbing lysine-rich intrinsically disordered regions (IDRs) in yeast snoRNP proteins on the nucleolar structure/function relationship. Having identified these IDRs in a subset of early-acting ribosome biogenesis factors, and giving the propensity of IDRs to self-associate and undergo liquid liquid phase shifts (LLPSs) that have been implicated in the formation of membraneless organelles such as the nucleolus, the authors propose that these associations play both a structural and functional role in ribosome biogenesis. A series of systematic and generally well-controlled experiments provides evidence to support this hypothesis. They include demonstration of the ability of KKE/D domains to self-associate both in vitro (in the presence of poly-U RNAs) and in vivo (homo- and heterotypic Y2H interactions). The impact of multiple KKE/D deletions in multiple snoRNP proteins on cell growth and pre-rRNA processing was then assessed, followed by imaging, ChIP-qPCR, Northern blot, co-IP and BioID assays to demonstrate the contribution of the KKE/D domain to the recruitment of snoRNP proteins to the vicinity of actively transcribing rDNA genes. Lastly, the impact of these mutations on nucleolar compaction in response to stress (in this case, TORC1 inactivation by rapamycin treatment) was assessed using imaging-based assays. The authors then present a model in which these KKE/D domains in multiple snoRNP proteins promote electrostatic-based association with nascent pre-mRNAs to enhance their activities under normal growth conditions, shifting to self-association under stress conditions to promote formation of the subnucleolar condensates that sequester ribosome biogenesis factors until the stress is removed.

I think that this work will be highly significant to the field, as the contribution of LLPS to nucleolar formation and spatial partitioning is hotly debated. Indeed, a major strength here is the authors' acknowledgment and discussion of this current debate in the field. One of the most striking results presented here is the EM image in Fig. 3F that suggest loss of the bipartite nucleolar substructure (ie phase-dense FC/DFC region appears to be gone) in the multiple KKE/D deletion mutant under standard growth conditions. Fluorescence images obtained in parallel show diffusion of the Cbf5 deletion mutant throughout the nucleolus in this background. What is missing, however, is a comprehensive assessment of the distribution of various FP-tagged wild type FC/DFC and GC factors in the nucleolus in this $\Delta\Delta\Delta\text{kk}$ background under standard growth conditions. Given that their EM data suggest that formation of a distinct FC/DFC condensate has been compromised in these mutants, and that the contribution of LLPS to formation of a bipartite nucleolus is under debate in the field, I think it is important to follow up this result. If they did, and the results did not support a model in which KKE/D domains impact partitioning, then this should be noted. Instead, the authors carefully delineate their model to claim an impact on recruitment to nascent pre-rRNAs under normal growth conditions and an impact on subnucleolar compaction under stress conditions. While I appreciate that they do not want to risk over-interpreting their results, they have presented data that suggests that these domains may contribute to LLPS-mediated subnucleolar partitioning under normal growth conditions and should therefore explore that further (ie assess the subnucleolar distribution of several DFC/FC and GC factors in unstressed conditions).

With respect to stress, the loss of rapamycin-induced condensate formation in the $\Delta\Delta\Delta\text{kk}$ cells is demonstrated both by the loss of FC/DFC factor accumulation in a condensate and loss of GC factor (Enp1) exclusion from an FC/DFC factor condensate (Fig. 7). This Enp1 result is intriguing (suggests that nucleolar substructure is selectively impacted), but could be further strengthened by testing additional GC factors such as Fkbp39/41. Recent work by another group (Ugolinbi et al Mol Cell 2022) suggests that these proteins (yeast homologues of mammalian nucleophosmin) contribute to the organization of ribosome biogenesis by partitioning nascent 60S subunits away from chromatin and nascent 40S subunits via LLPS. They are therefore obvious GC factors to test to see if/how they are impacted (in both stressed and unstressed cells) by the

changes in nucleolar substructure induced by the $\Delta\Delta\Delta$ kk mutations in snoRNP proteins.

Assessment of rRNA pseudouridylation and methylation in the KKE/D deletion mutants in Fig. 2C-F was carried out with 3 biological replicates and statistical analysis, however the impact on growth is shown as a single growth assay (Fig. 2A) and the impact on pre-mRNA processing as a single Northern blot (Fig. 2B), with a cropped replicate included as Fig. S2. In the text (lines 170-171) the authors claim a “significant growth delay” and a “mild early processing defect”. To support these claims, these assays should be repeated (n of at least 3) and quantified for statistical analysis.

The authors also need to clarify their biological replicates for all experiments that are presented here. For example, for the imaging-based assays (eg Fig. 3C) they note that >100 cells were assessed. Is that 100 technical replicates in 1 biological replicate, or 100 technical replicates collected across 3 biological replicates (ie individual experiments)? And for the n values in their statistical analyses, did they use n = 3 (or however many biological replicates they carried out) or n = 100 (which would provide a falsely low p value)? It is difficult to assess the underlying variability from the box plots that are presented (a colour-coded beeswarm or violin plot would be more effective).

Minor:

The introduction provides a very nice overview of mammalian tripartite nucleolar substructure (FC/DFC/GC), but to clarify the results of this study for readers not as familiar with yeast morphology the authors should note that yeast nucleoli are crescent-shaped and have a bipartite substructure (chromatin-associated FC/DFC region and GC region).

Reviewer #2

(Remarks to the Author)

This impressively detailed and extensive study documents multiple features of the biology of snoRNP proteins that are linked to their lysine-rich KKE/D domains. The possible biological significance of these domains – that are characteristic of snoRNP proteins (and others) – certainly merits investigation.

It is especially notable that these domains promote protein association with ribosomal DNA.

Rather than providing a coherent story, however, the text seems to be an enumeration of observations. It would help the reader if the authors began with a single summary statement or hypothesis that pulls the rest of the article together. Perhaps the successive sections should each start with a question. Perhaps more of the descriptive data should be supplemental.

The observations on rRNA processing “defects” in strains lacking multiple KKE/D domains (Figure 2B) are not convincing.

The text says that KKE/D truncations “abolish the visualization of DFCs,” but the putative DFCs are not even labeled in the EMs in Figure 3F – so it is impossible to judge what is meant.

In lines 265/266 we are told to consider the possibility that KKE/D domains interact directly with nascent ribosomal RNA. When we get to lines 311/312 we learn that interactions between these domains are not responsible for interactions with nascent rRNA. These two statements and the corresponding data should be brought together, perhaps at the beginning (lines 265/266).

In line 319 we are told that when ribosomal RNA is driven by a pol II promoter, the corresponding genes are not embedded in the nucleolus. Please cite a reference if this is true.

It is curious that the so-called “nucleolar compaction” - that the authors see in normal cells treated with rapamycin – depends on the KKE/D domains. But this compaction seems to be a weak point for further investigation since its fundamental significance is unclear.

The last sentence of the Abstract is not needed.

Lesser concerns:

The text sometimes uses the term “coacervate.” It is not clear whether the authors intend to distinguish these units from condensates.

The authors repeatedly refer to concentrations of a given protein as being a condensate. In order to do so, they need to define what they mean by a condensate and they should explain why the accumulations with which they are concerned should not simply be referred to as “a concentration” or “an aggregate.”

The Introduction mentions “conditional properties of IDRs” and their “functional plasticity.” Please explain what is meant.

Explain what is meant by a Kappa (K) index.

Despite the data of Wei et al. is it well-established that BMH-21 specifically inhibits pol 1 ?

In all CHIP figures, the IP/Input estimates are very low. Why is this ? Does it diminish the reliability of the data ?

The title on line 263/264 should be rewritten.

Reviewer #3

(Remarks to the Author)

The manuscript by Dominique et al address a very interesting novel finding and important scientific question in a constructive and clear manner, and put effort in describing their results in a critical and clear way. The story is coherent and the conclusions in general well justified. The paper describes a new function of the abundant, lysine-rich intrinsically disordered regions, that are present in snoRNPs and other early acting, nucleolar assembly and maturation factors. Using microscopy, IP and growth assays, the authors could underline an important role of these KKE/D domains in recruiting snoRNPs to the transcription sites of rDNA genes. Their results suggest that KKE/D domains are important to maintain the multilayers organization of the nucleolus by acting as ligand for LLPS in concert with GAR domain containing proteins. The manuscript is very well written, with clear illustrations and esthetic figures. We agree with most of the conclusions of this study, but have several comments that should be addressed.

Major points:

- (1) for the mutants, the authors make deletions of the KKE/D sequences, which also will significantly shorten the length of these IDRS, which besides the motif itself could also be affecting their function. It would be good to replace the IDRs with linkers of the same / similar length for at least some of the experiments.
- (2): Figure 1J: this figure is very zoomed in / misleading (different scale bars in pictures next to each other); the 'condensates' shown could also be aggregates, which start to look roundish at this magnification. A good control for this would be to show the dissolution experiment with NaCl as for the polyU condensates.
- (3) Figure 2B and Supplementary Figure 2: please quantify the Northern Blots, some of the statements from the text are not obvious
- (4) Figure 4: why was the entire C-terminal domain of Cbf5 - containing additional sequence elements - used for these experiments, not only the "KKE/D" region
- (5) Figure 4A/B: the authors should show the cellular localization of the GFP control. If it does not localise to the nucleus / nucleolus as the KKE-GFP, it would not be the appropriate control. Also the expression levels should be quantified (and ideally in a similar range)
- (6) Line 331 and further: the wording "Cbf5 condensate" is probably not yet appropriate at this stage, unless there is data not shown. It would be more correct to describe it for example as "Cbf5 granule" or 'recruitment' (line 339/376)
- (7) Figure 6A: please provide statistics how often perfect colocalization of Cbf5 and Net1-mKate occurs (only one cell visible)
- (8) Line 340 / figure 6D: exclusion is very hard to judge from these images, especially since it is very few cells. We rather would judge what we see as "mislocalisation". Please clarify and quantify
- (9) 6B/C: why do the authors only show images for single (Nop56- Δ kk), not double knockout ($\Delta\Delta$ kk) strains? From the quantification in Fig. 6C, the pictures were available. Please also be consistent in the text

Minor points:

- (10) Supplementary table 1:
 - NOP56 is duplicated; YOR310C is NOP58
 - some of these proteins have more than one IDR of >30 AA; it is not clear if they count them 'all together'?
- (11) Supplementary figure 1A: are the names of the proteins in the different organisms in S1A all the same? Please give the names of the protein for each species, and the start / end residues of the domains
- (12) Figure 2C/E/G: please indicate in the figure legend the function of the blue shaded area
- (13) Figure D/F: **** is not described in the statistical analysis section of the methods part
- (14) Figure 3B and others: red / green is not ideal for color-blind people
- (15) Figure 3B/C: B is Nop1-mKate, the quantification in C is Nop1-mCherry (the pictures are in the supplement). Please clarify and be consistent.
- (16) Figure 3E: the methods section for the qPCR is missing (experimental protocol, quantification, normalization).

Therefore, it is very hard to understand and judge how the scale (IP/input) in 3E was calculated.

(17) Figure 3F: please add arrows / labels and / or a schematic representation to guide the reader what to look at

(18) Figure 3A / 4A: please clarify the schemes; the KKE / KKE/D domain seems to be differently defined (e.g. the arrows Figure 3 vs Figure 4 (402-483 vs 392-483), and clarify what was used in which experiment.

(19) S6B, pictures of post-diauxic stress are not mentioned or used in the paper

(20) please pay attention to nomenclature writing (genes vs. proteins vs. genomic locus / mutants etc), it is not always consistent / uniform (e.g. Figure 3B labelling of figures vs. Figure legend)

Typos etc:

- Lines 136 and 139: 'hereafter called the KKE/D domain' is duplicated
- Table S2: nomenclature of double mutants not uniform (e.g. $cbf5-\Delta kk$ versus $cbf5-\Delta \Delta k$ versus)
- line 330: check punctuation (we suspect there should be a comma)
- line 309: please spell out what the $cbf5-\Delta kk$ mutant is somewhere in the text

Version 1:

Reviewer comments:

Reviewer #1

(Remarks to the Author)

In my initial review I noted that I felt the work would be highly significant to the field, as the contribution of LLPS to nucleolar formation and spatial partitioning is hotly debated. That said, there were several issues raised by me and the other reviewers. I feel that the revised manuscript has been strengthened by data added to address these comments, including additional CLEM images, assessment of additional nucleolar factors and demonstration of the statistical significance of the growth assay and Northern blot results. The text revisions, which addressed both minor (grammar, consistency) and major (clarification of certain methods and results) issues, have also improved readability.

Reviewer #2

(Remarks to the Author)

This revised manuscript includes detailed responses to the many questions posed by the reviewers. In that regard, the job is done; however, neither the title nor the abstract announces one key take-home message and its significance. If the authors can modify these items they will heighten the impact of their findings.

Reviewer #3

(Remarks to the Author)

The authors have addressed my comments and put together a beautiful manuscript.

There is only one point / figure that needs to be clarified further, which is 1 I/J. For condensates in and higher salt, the authors alter in the description between "sensitive to the addition" and "in the presence of" (the latter being more likely when reading the methods part). It is unfortunately important whether the condensates were formed in the presence of high salt, or whether they were first formed in low salt and then dissolved by addition of 300 mM salt, which would demonstrate reversibility, distinguishing them from aggregates, and thus what I would suggest should be the goal of this experimental setup. Therefore, I would suggest to repeat this experiment accordingly (i.e. first make condensates, then add high salt). Furthermore, there are some aggregates visible in 1J high salt in the brightfield, but not in the GFP channel - could the authors please briefly explain this discrepancy?

RESPONSE TO REVIEWERS' COMMENTS

Reviewer #1

This study presents a systematic analysis of the downstream effects of perturbing lysine-rich intrinsically disordered regions (IDRs) in yeast snoRNP proteins on the nucleolar structure/function relationship. Having identified these IDRs in a subset of early-acting ribosome biogenesis factors, and giving the propensity of IDRs to self-associate and undergo liquid liquid phase shifts (LLPSs) that have been implicated in the formation of membraneless organelles such as the nucleolus, the authors propose that these associations play both a structural and functional role in ribosome biogenesis. A series of systematic and generally well-controlled experiments provides evidence to support this hypothesis. They include demonstration of the ability of KKE/D domains to self-associate both in vitro (in the presence of poly-U RNAs) and in vivo (homo- and heterotypic Y2H interactions). The impact of multiple KKE/D deletions in multiple snoRNP proteins on cell growth and pre-rRNA processing was then assessed, followed by imaging, ChIP-qPCR, Northern blot, co-IP and BioID assays to demonstrate the contribution of the KKE/D domain to the recruitment of snoRNP proteins to the vicinity of actively transcribing rDNA genes. Lastly, the impact of these mutations on nucleolar compaction in response to stress (in this case, TORC1 inactivation by rapamycin treatment) was assessed using imaging-based assays. The authors then present a model in which these KKE/D domains in multiple snoRNP proteins promote electrostatic-based association with nascent pre-mRNAs to enhance their activities under normal growth conditions, shifting to self-association under stress conditions to promote formation of the subnucleolar condensates that sequester ribosome biogenesis factors until the stress is removed.

I think that this work will be highly significant to the field, as the contribution of LLPS to nucleolar formation and spatial partitioning is hotly debated. Indeed, a major strength here is the authors' acknowledgment and discussion of this current debate in the field. One of the most striking results presented here is the EM image in Fig. 3F that suggest loss of the bipartite nucleolar substructure (ie phase-dense FC/DFC region appears to be gone) in the multiple KKE/D deletion mutant under standard growth conditions. Fluorescence images obtained in parallel show diffusion of the Cbf5 deletion mutant throughout the nucleolus in this background.

What is missing, however, is a comprehensive assessment of the distribution of various FP-tagged wild type FC/DFC and GC factors in the nucleolus in this $\Delta\Delta\Delta$ kk background under standard growth conditions. Given that their EM data suggest that formation of a distinct FC/DFC condensate has been compromised in these mutants, and that the contribution of LLPS to formation of a bipartite nucleolus is under debate in the field, I think it is important to follow up this result. If they did, and the results did not support a model in which KKE/D domains impact partitioning, then this should be noted.

Instead, the authors carefully delineate their model to claim an impact on recruitment to nascent pre-rRNAs under normal growth conditions and an impact on subnucleolar compaction under stress conditions. While I appreciate that they do not want to risk over-interpreting their results, they have presented data that suggests that these domains may contribute to LLPS-mediated subnucleolar partitioning under normal growth conditions and should therefore

explore that further (ie assess the subnucleolar distribution of several DFC/FC and GC factors in unstressed conditions).

*With respect to stress, the loss of rapamycin-induced condensate formation in the $\Delta\Delta\Delta$ kk cells is demonstrated both by the loss of FC/DFC factor accumulation in a condensate and loss of GC factor (*Enp1*) exclusion from an FC/DFC factor condensate (Fig. 7). This *Enp1* result is intriguing (suggests that nucleolar substructure is selectively impacted), but could be further strengthened by testing additional GC factors such as *Fkbp39/41*. Recent work by another group (Ugolinbi et al Mol Cell 2022) suggests that these proteins (yeast homologues of mammalian nucleophosmin) contribute to the organization of ribosome biogenesis by partitioning nascent 60S subunits away from chromatin and nascent 40S subunits via LLPS. They are therefore obvious GC factors to test to see if/how they are impacted (in both stressed and unstressed cells) by the changes in nucleolar substructure induced by the $\Delta\Delta\Delta$ kk mutations in snoRNP proteins*

We first thank Reviewer #1 for these positive comments, and we agree that it will be interesting to explore further to what extent KKE/D domain truncations impact nucleolar organization. As described in more detail in the response to Reviewer #2 below, we have added new cryo-EM data to confirm that KKE/D domain truncations alter nucleolar morphology during exponential growth (**New Supplemental Figure S3F**).

Furthermore, as requested by Reviewer #1, we also examined in the KKE/D domain mutant strain the localization of several ribosome biogenesis factors representative of different groups of nucleolar proteins that have been classified in a recent study by Alan Tartakoff according to their position relative to snoRNPs in both physiological and stress conditions (Tartakoff et al., 2021). Three main categories emerged from this latter study: proteins that colocalize constitutively with snoRNPs in both physiological and stress conditions, such as *Nsr1*; proteins that colocalize with snoRNPs only during exponential growth and relocate to the nucleolar outer layer after stress; proteins enriched in late ribosome biogenesis factors that never colocalize with snoRNPs during exponential growth or stress.

We therefore decided to label *Nsr1* as a representative of a constitutive component of the inner nucleolar layer, similar to snoRNPs. *Utp22*, *Utp25* and *Rrp5* were used as representatives of proteins that relocate from the inner to the outer nucleolar layer during stress. *Enp1*, *Fpr3* and *Rrp1* were chosen as constitutive components of the outer nucleolar layer (**New Supplemental Figures S7A, S7B**). We also wanted to mention that unfortunately, we could not test the localization of several other late ribosome biogenesis factors known to accumulate in the GC (*Nog2*, *Enp2*, *Rrp6*, *Nog1*, *Nop4*), as their GFP-tagged version caused substantial growth defects in combination with the KKE/D domain mutation. To avoid misinterpretation, we decided to remove them from this analysis, which consists in assessing the capacity of a given nucleolar protein to accumulate in the nucleolus in wild-type or KKE/D domain mutant cells in different growth conditions.

Interestingly, we now show that the nucleolar accumulation of *Nsr1* and *Rrp5* is affected by the truncation of the KKE/D domains during exponential growth, whereas the localization of other early or late factors is not (**New Supplemental Figures S7A, S7B**). During stress, *Rrp5* accumulation in the compacted nucleolus is not significantly affected, nor is that of other factors tested, in contrast to *Nsr1*. *Nsr1* has been reported to be a constitutive component of this inner nucleolar layer enriched in snoRNPs and our new data confirmed that the KKE/D domains are essential for the formation of this subnucleolar domain and the sequestration of a specific pool

of abundant nucleolar proteins including snoRNPs, RNAPI and other early ribosome biogenesis factors such as Nsr1.

In this analysis, we only used the intensity of the nucleolar GFP signal adjacent to rDNA (detected by Net1) as a proxy for correct nucleolar accumulation, but our analysis remains superficial and does not rule out that the precise localization of proteins inside the nucleolus may be affected. Future experiments will be required to clarify the differential dependence on the KKE/D domains in the correct targeting of these nucleolar proteins, but we believe that the results shown in the revised manuscript seem sufficient to confirm our data presented in the first version, suggesting that the nucleolar substructure is selectively affected by KKE/D domain truncations. It will be of particular interest to understand why the localization of Nsr1 or Rrp5, but not that of UTP complex components, is affected in the KKE/D domain truncation mutant strain during exponential growth. The fact that the UTP components were proposed to be recruited independently of U3-snoRNP and that Nsr1 contains a GAR domain similar to snoRNPs may explain this different dependence on the KKE/D domains.

We presented and commented these results in our revised manuscript (**Lines 407-425**) without going into too many details due to length limitations.

Assessment of rRNA pseudouridylation and methylation in the KKE/D deletion mutants in Fig. 2C-F was carried out with 3 biological replicates and statistical analysis, however the impact on growth is shown as a single growth assay (Fig. 2A) and the impact on pre-mRNA processing as a single Northern blot (Fig. 2B), with a cropped replicate included as Fig. S2. In the text (lines 170-171) the authors claim a “significant growth delay” and a “mild early processing defect”.

To support these claims, these assays should be repeated (n of at least 3) and quantified for statistical analysis.

As suggested by Reviewer #1, we have now included biological replicates of all growth assays (**New Supplemental Figure S2A**), and Northern blot experiments (**New Supplemental Figures S2C, S2D, S2E**) to support our claims. We also added the doubling time of all mutant strains measured in liquid cultures at 30°C and 37°C, to provide a more accurate view of the growth defect associated with each mutation (**New Supplemental Figure S2B**). These new data confirmed our initial observations.

The authors also need to clarify their biological replicates for all experiments that are presented here. For example, for the imaging-based assays (eg Fig. 3C) they note that >100 cells were assessed. Is that 100 technical replicates in 1 biological replicate, or 100 technical replicates collected across 3 biological replicates (ie individual experiments)? And for the n values in their statistical analyses, did they use n = 3 (or however many biological replicates they carried out) or n = 100 (which would provide a falsely low p value)? It is difficult to assess the underlying variability from the box plots that are presented (a colour-coded beeswarm or violin plot would be more effective).

We agree with Reviewer #1 that the text was too concise and lacked important information regarding our biological replicates. We have expanded this part of the Materials and Methods section. P values were calculated using unpaired two-samples Wilcoxon tests and n indicates the number of cells analyzed (>100), pooled from images from at least three independent biological replicates. We added this information in the “Fluorescence microscopy, quantifications, and statistical analyses” section (**Lines 637-645**). In addition, we now present

most of the quantifications as violin plots as requested by Reviewer #1 (**New Figures 3C, 6C, 6G, 7C, 7D, S3C, S6A, S6G, S6I, S6O, S7B**).

Minor:

The introduction provides a very nice overview of mammalian tripartite nucleolar substructure (FC/DFC/GC), but to clarify the results of this study for readers not as familiar with yeast morphology the authors should note that yeast nucleoli are crescent-shaped and have a bipartite substructure (chromatin-associated FC/DFC region and GC region).

We really appreciate this comment on our introduction, but we would like to mention that there is still some debate about the tripartite or bipartite structure of the yeast nucleolus, and we currently have some unpublished new data supporting that multiple subcompartments can be distinguished even in a yeast nucleolus. For example, in **Figure 3F**, careful examination of the Cbf5-GFP and Net1-mKate signals allows to delineate two distinct subnucleolar compartments that might correspond to DFC and FC, respectively, as suggested by older studies (Léger-Silvestre et al, 1999). Similarly, recent studies in yeast distinguished distinct localizations of the rDNA, snoRNPs and late ribosome biogenesis factors (Tartakoff et al., 2021). We are currently attempting to validate these observations in yeast using high-resolution microscopy, and we think it might be confusing to mention this debate in the present study. We hope Reviewer #1 will understand this point.

Reviewer #2

This impressively detailed and extensive study documents multiple features of the biology of snoRNP proteins that are linked to their lysine-rich KKE/D domains. The possible biological significance of these domains – that are characteristic of snoRNP proteins (and others) – certainly merits investigation.

It is especially notable that these domains promote protein association with ribosomal DNA.

We thank Reviewer #2 for these positive comments.

Rather than providing a coherent story, however, the text seems to be an enumeration of observations. It would help the reader if the authors began with a single summary statement or hypothesis that pulls the rest of the article together. Perhaps the successive sections should each start with a question. Perhaps more of the descriptive data should be supplemental.

We are sorry that Reviewer #2 did not appreciate the general organization and writing of our story, in contrast to Reviewer #3 ("*the story is coherent and the conclusions are generally well justified*" and "*the manuscript is very well written*"). Regarding the “enumeration of observations” pointed out by Reviewer #2, we must admit that we have deliberately chosen to remain (too?) descriptive in many parts of the study to avoid over-interpretation of the data, which has recently led to numerous conflicting studies and over-simplifications in the field of condensate formation. Nevertheless, we have tried to improve the readability of our revised manuscript, for example by rephrasing the beginning of the paragraph starting **Line 324**. We also added a new paragraph in the last part of the result section (**Lines 426-443**) to separate the section that specifically addresses the link between RNAPI activity and snoRNP condensation from the part describing the consequences of KKE/D domain truncation on nucleolar organization.

The observations on rRNA processing “defects” in strains lacking multiple KKE/D domains (Figure 2B) are not convincing.

A similar concern has been raised by Reviewer #1 and as pointed out above, we have now included biological replicates and quantification of these Northern blot data in the revised version of the manuscript (**New Supplemental Figure S2**).

The text says that KKE/D truncations “abolish the visualization of DFCs,” but the putative DFCs are not even labeled in the EMs in Figure 3F – so it is impossible to judge what is meant.

As requested by Reviewer #2, we have delimited the dense area corresponding to the DFC in the wild-type strain (**new Figure 3F**). In the KKE/D domain mutant, given the lack of evidence for a dense area in all observed nuclei, it was not possible to identify and then delineate accurately a DFC, (or even a nucleolus). To confirm our claim, and to go one step further in characterizing the nucleolus of the KKE/D domain mutant cells, we performed cryo-electron microscopy analyses using CLEM (Correlative Light and Electron Microscopy), which allowed us to directly visualize Net1 (a protein associated with rDNA) in the cryo-sections observed by electron microscopy. This approach allowed us to confirm that our cryo-sections were located in nuclear sections corresponding to the nucleolus in both wild-type and mutants nuclei (**New Supplemental Figure S3F**). Using this method, we confirmed our primary observation that the classical dense area expected to localize around the Net1 signal in a wild-type strain is not observed in the KKE/D domain mutant. These data confirmed our initial claim that the KKE/D domain truncation has a direct effect on the organization of the DFC.

In lines 265/266 we are told to consider the possibility that KKE/D domains interact directly with nascent ribosomal RNA. When we get to lines 311/312 we learn that interactions between these domains are not responsible for interactions with nascent rRNA. These two statements and the corresponding data should be brought together, perhaps at the beginning (lines 265/266).

We agree with Reviewer #2 that the beginning of this section is confusing. As suggested, we clarified and rephrased these different sentences (**Line 324**). Our data support the notion that the KKE/D region of Cbf5 is sufficient and required for interaction with nascent rRNAs, and further indicate that the multiple interactions between KKE/D domains do not seem to be required for this interaction with nascent rRNAs, but for compaction of snoRNPs during stress.

In line 319 we are told that when ribosomal RNA is driven by a pol II promoter, the corresponding genes are not embedded in the nucleolus. Please cite a reference if this is true.

In the original manuscript, line 319, we wanted to highlight that a yeast two-hybrid assay allows to address the KKE/D domain interaction properties outside of the nucleolus, as the *HIS3* and *ADE2* reporter genes, whose transcription is under the control of the *GAL1* and *GAL2* promoter, respectively, are not embedded in rDNA regions and are therefore not expected to be transcribed in the nucleolus near the nascent rRNA. This sentence has now been corrected (**Lines 327-329**).

It is curious that the so-called “nucleolar compaction” - that the authors see in normal cells treated with rapamycin – depends on the KKE/D domains. But this compaction seems to be a weak point for further investigation since its fundamental significance is unclear.

Independently from its role in ribosome biogenesis, it is becoming increasingly clear that the nucleolus is an important membraneless organelle involved in stress adaptation, with nucleolar sequestration of regulatory and toxic proteins, allowing their refolding and stress adaptation (Frottin et al., 2019; Grummt, 2013). Similarly, in yeast, nucleolus compaction has been shown to be important for stress adaptation (Mostofa et al., 2019, Perdomo and Machin 2019) and we have preliminary data that seem to indicate that truncation of KKE/D domains reduces viability of cells following long-term exposure to stress. In the future, we plan to directly test the role of nucleolar compaction and KKE/D domains in maintaining a functional proteome under different stress conditions. Nucleolar compaction may also be an important phenomenon during cell division and we believe that the KKE/D domains may be directly involved in this process through their ability to act as scaffolds maintaining critical ribosome biogenesis factors in a dedicated subnuclear compartment.

The last sentence of the Abstract is not needed.

We would like to keep this concluding sentence to place the data in a wider context.

Lesser concerns:

The text sometimes uses the term “coacervate.” It is not clear whether the authors intend to distinguish these units from condensates.

The authors repeatedly refer to concentrations of a given protein as being a condensate. In order to do so, they need to define what they mean by a condensate and they should explain why the accumulations with which they are concerned should not simply be referred to as “a concentration” or “an aggregate.”

We understand these comments of Reviewer #2, as there are indeed different definitions and uses of the term condensate in the field. In the new version of our manuscript, we have defined the term “condensate” in the Introduction section (Lines 68-70). We have defined a condensate as a dynamic subcellular membraneless compartment that promotes local accumulation of given proteins at high concentration. One of the main differences between protein aggregates and condensates is their reversibility, as an aggregate is generally considered as an irreversible structure. Finally, the term “coacervate” is more often associated with the LLPS mechanisms, whereas “condensate” is a more general term and does not refer to the underlying mechanism of formation.

For these reasons, we have only used the term “coacervate” in Figure 1, as the observed droplets correspond to the general definition of *in vitro* coacervates, driven by multivalent associations between peptides and RNAs. We decided to use the term “condensate” for the other part of the study because condensate formation could be the result of an equilibrium between different biophysical processes, including LLPS, PPPS (polymer-polymer phase separation), percolation or interactions with intercluster binding sites (ICBS) without phase separation, and specific tools will be required to delineate the involvement of each process.

The Introduction mentions “conditional properties of IDRs” and their “functional plasticity.” Please explain what is meant. Explain what is meant by a Kappa (K) index.

We now added two sentences describing in more detail the Kappa index (Lines 123-126) and what we called “functional plasticity” (Line 74).

Despite the data of Wei et al. is it well-established that BMH-21 specifically inhibits pol 1 ?

There are now several studies (Colis et al., 2014; Jacobs et al., 2022; Peltonen et al., 2014a, 2014b, Azouzi et al., 2024) showing that BMH21 has a specific role on RNAPI activity.

In all CHIP figures, the IP/Input estimates are very low. Why is this ? Does it diminish the reliability of the data ?

Most rDNA ChIP experiments in the literature have been performed for proteins that directly bind to rDNA, contrary to the proteins in our study that are very likely cross-linked to rDNA indirectly through their association with the nascent transcripts. This may explain why our IP/Input ratios are generally lower. Our results are nevertheless consistent with several ChIP experiments performed in other studies, for example a recent ChIP using as bait Nhp2 (a component of H/ACA snoRNPs) that reported IP/Input ratios in a similar range to those obtained for Cbf5 or the KKE/D-GFP construct (PMID: 34409714, Figure 3). This suggests that our IP/Input ratios are comparable to those obtained for this type of proteins in other laboratories.

The title on line 263/264 should be rewritten.

As suggested, we have changed the title to “*Efficient targeting of the snoRNP KKE/D domains to transcribed rDNA genes depends on rRNA production.*”, **lines 271-272** of the revised manuscript.

Reviewer #3:

The manuscript by Dominique et al address a very interesting novel finding and important scientific question in a constructive and clear manner and put effort in describing their results in a critical and clear way. The story is coherent and the conclusions in general well justified. The paper describes a new function of the abundant, lysine-rich intrinsically disordered regions, that are present in snoRNPs and other early acting, nucleolar assembly and maturation factors. Using microscopy, IP and growth assays, the authors could underline an important role of these KKE/D domains in recruiting snoRNPs to the transcription sites of rDNA genes. Their results suggest that KKE/D domains are important to maintain the multilayers organization of the nucleolus by acting as ligand for LLPS in concert with GAR domain containing proteins. The manuscript is very well written, with clear illustrations and esthetic figures. We agree with most of the conclusions of this study, but have several comments that should be addressed.

We first thank Reviewer #3 for these highly positive comments.

Major points:

(1) for the mutants, the authors make deletions of the KKE/D sequences, which also will significantly shorten the length of these IDRS, which besides the motif itself could also be affecting their function. It would be good to replace the IDRs with linkers of the same / similar length for at least some of the experiments.

We fully agree that the length of an IDR is an important point that we did not discuss in the initial version of the manuscript. IDR length has indeed been reported to influence their flexibility and conformational entropy. These two parameters are critical for the folding and functionality of adjacent domains when IDRs act as linkers, whereas when they are localized at the N- or C-terminus, as for KKE/D domains, the IDR length mainly influences its interaction capacity by changing the radius of gyration (in a manner also dependent on the amino acid composition). Importantly, it has been reported that different characteristics of IDRs (length, repeated motif, amino acid arrangement in the sequence) are fundamentally intertwined (PMID: 38297184), which explains why it is difficult to distinguish, upon IDR truncation, between the specific effect of IDR length reduction and the effect of a specific amino acid motif truncation. The specific conservation of the IDRs of Cbf5, enriched in lysine blocks, in the eukaryotic lineage, strongly suggests that any IDR of similar size as the KKE/D domain should not be sufficient to replace the function of this lysine-rich extension. To confirm this hypothesis, and as requested by Reviewer #3, we mutated the KKE/D domain of Cbf5 by replacing most of its charged residues with uncharged amino acids (KKEmut IDR). The KKEmut IDR amino acid sequence was first designed *in silico* by mutating charged amino acids (mainly K and E residues). The choice of mutated amino acids has been done in order to i/ abolish the charged nature of the KKE/D domain, ii/ maintain its flexibility and size, iii/ avoid non-physiological large repetitions of similar amino acids that could affect translation and/or aggregation. This sequence was then codon-optimized for expression in yeast and synthesized by Eurofins. We also considered an alternative approach by replacing the KKE/D domain of Cbf5 with the polyampholyte IDR of the nucleolar protein Enp1, which is negatively charged contrarily to the KKE/D domain (**New Supplemental Figure S6J**). These new domains of similar size are predicted *in silico* to be disordered but, as expected, they show drastically different properties as illustrated on a phase diagram (**New Supplemental Figure S6K**).

In vivo, these modified Cbf5 proteins are expressed at their expected size (**New Supplemental Figure S6M**) but modifying the amino acid composition of the KKE/D domain or replacing this domain by the Enp1 IDR more drastically affected cell growth compared to its truncation (**New Supplemental Figure S6L**). Moreover, nucleolar accumulation of these Cbf5 mutants both in exponentially growing and stressed cells was also strongly affected (**New Supplemental Figure S6N, S6O**). These new data demonstrate that any IDR of similar size is not sufficient to functionally substitute for the KKE/D domain. It might be interesting in future experiments to increase or decrease the length of the KKE/D domain to understand specifically the importance of its size. We added the description of these new experiments in the revised manuscript (**Lines 372-385**).

(2): *Figure 1J: this figure is very zoomed in / misleading (different scale bars in pictures next to each other); the 'condensates' shown could also be aggregates, which start to look roundish at this magnification. A good control for this would be to show the dissolution experiment with NaCl as for the polyU condensates.*

As requested by Reviewer #3, we have reorganized this figure (**New Figure 1J**). In addition, we show that these round-shaped structures observed in the presence of total RNA are also sensitive to NaCl (**New Figure 1J**).

(3) *Figure 2B and Supplementary Figure 2: please quantify the Northern Blots, some of the statements from the text are not obvious*

We added replicates and quantification, confirming a slight but reproducible effect of KKE/D domain mutations on pre-rRNA processing (**New Supplemental Figure S2C, S2D, S2E**).

(4) Figure 4: why was the entire C-terminal domain of Cbf5 - containing additional sequence elements - used for these experiments, not only the “KKE/D” region

This is another good point raised by Reviewer #3. Although the lysine repeats are the prominent feature of the KKE/D domain, it is important to note that these repeats are embedded in a larger IDR and, as explained above, the length of the IDR is an important parameter defining, among other things, its radius of gyration. Therefore, for the loss-of-function mutant, we chose to generate a truncation removing specifically the lysine-rich region of Cbf5, starting at the first lysine (position 403) and keeping the beginning of the IDR to decrease the potential impact of the truncation on the adjacent domain of Cbf5. In contrast, we retained for the gain-of-function approach the larger IDR sequence (392-483) fused to GFP to conserve the general properties of this KKE/D domain. We have modified the schematic representation of Cbf5 in **Figures 3A, 4A** and added the construction coordinates in the main text as well as in the legend to make it easier to understand which part of Cbf5 is being studied.

(5) Figure 4A/B: the authors should show the cellular localization of the GFP control. If it does not localise to the nucleus / nucleolus as the KKE-GFP, it would not be the appropriate control. Also the expression levels should be quantified (and ideally in a similar range)

As requested by Reviewer #3, we localized and quantified the expression levels of GFP and KKE/D-GFP (**New Supplemental Figures S4A, S4B**) that are expressed under the control of a similar strong promoter (pGARI). GFP alone reproducibly accumulated at higher levels than KKE/D-GFP and localized as a diffuse signal in the cytoplasm, nucleus and nucleolus (**New Supplemental Figure S4A**). We agree with the Reviewer #3 that the different level of nucleolar accumulation of GFP and KKE-GFP has to be considered in our interpretation of the KKE/D-GFP ChIP experiment. Nevertheless, we would like to insist on the fact that this experiment has to be connected to the other part of the manuscript showing in **Figure 3E** that Cbf5 lacking its KKE/D domain remains in the nucleolus but does not as efficiently immunoprecipitate rDNA and modify rRNAs as full length Cbf5. In regard to this “loss-of-function” experiment, we then sought in **Figure 4** to undertake a complementary, “gain-of-function” approach to confirm this interpretation, by comparing the properties of a neutral GFP domain fused or not to a KKE/D domain. For this reason, we systematically compared KKE/D-GFP with GFP alone in these experiments, and despite the relevant limitations pointed out by the reviewer, we believe that this approach is highly complementary to other observations made in this manuscript and although it is not sufficient *per se*, it does support the intrinsic ability of the KKE/D domain to facilitate access in close proximity to rDNA. We have changed the text to emphasize that the result of this ChIP needs to be linked to other experiments (**Line 248**).

(6) Line 331 and further: the wording “Cbf5 condensate” is probably not yet appropriate at this stage, unless there is data not shown. It would be more correct to describe it for example as “Cbf5 granule” or ‘recruitment’ (line 339/376)

The “condensate” terminology has already been used to describe snoRNP proteins or rDNA punctate structures in the nucleolus (Lawrimore et al, 2024), suggesting that it might be suitable to use this term. Nevertheless, as explained above to Reviewer #2, we agree that this point needed to be clarified and we added a general definition of what we call “condensate” in this study (**Line 68**).

(7) *Figure 6A: please provide statistics how often perfect colocalization of Cbf5 and Net1-mKate occurs (only one cell visible)*

Colocalization of Cbf5 and Net1-mKate is the most representative situation observed in virtually all the cells, explaining why we did not add statistics. To address this point, we simply provided an extended view of the field with other cells adjacent to those selected in the initial pictures, the large majority of which show the same localization profiles (**New Supplemental Figure S6B**).

(8) *Line 340 / figure 6D: exclusion is very hard to judge from these images, especially since it is very few cells. We rather would judge what we see as “mislocalisation”. Please clarify and quantify*

We have now quantified the number of cells in which we can observe such a fluorescence pattern (**New Supplemental Figure S6E**), which is characterized by the appearance of a circular structure with a low fluorescence signal at its center, suggesting that proteins tagged with GFP are excluded from this specific area. However, as suggested by Reviewer #3, the term "excluded" may not be appropriate and a specific set of experiments should be required to characterize such a physical mechanism. Therefore, we decided to replace "excluded" with "depleted".

(9) *6B/C: why do the authors only show images for single (Nop56- Δ kk), not double knockout ($\Delta\Delta$ kk) strains? From the quantification in Fig. 6C, the pictures were available. Please also be consistent in the text*

We now provide these images in **Figure 6B** (right panel for the double knockout).

Minor points:

(10) *Supplementary table 1:*

- NOP56 is duplicated; YOR310C is NOP58- some of these proteins have more than one IDR of >30 AA; it is not clear if they count them ‘all together’?

We thank Reviewer #3 for pointing out this mistake. In **Supplemental Table 1**, we indeed made some choices to represent both the individual amount of each protein and the diversity of the IDR proteome. We only kept large IDRs (>30 AA) and we only considered the largest IDR for a given protein. We have corrected the table and added the explanation requested by Reviewer #3.

(11) *Supplementary figure 1A: are the names of the proteins in the different organisms in S1A all the same? Please give the names of the protein for each species, and the start / end residues of the domains*

The name of the protein (Cbf5) is similar in fungi and plants but the orthologous protein in animals is called DKC1. The names are different in protists. We have added this information in the **Supplemental Figure S1A** or in the corresponding legend, as well as the start/end residues of the domains as requested by Reviewer #3.

(12) *Figure 2C/E/G: please indicate in the figure legend the function of the blue shaded area*

The blue shaded area allows visualization of all sites that are highly modified (HPS or RMS scores > 0.8). We have added this information to the figure legend.

(15) Figure 3B/C: B is Nop1-mKate, the quantification in C is Nop1-mCherry (the pictures are in the supplement). Please clarify and be consistent.

The quantification in **Figure 3C** is Nop1-mCherry, we clarified this point in the legend.

*(13) Figure D/F: **** is not described in the statistical analysis section of the methods part
(20) please pay attention to nomenclature writing (genes vs. proteins vs. genomic locus / mutants etc), it is not always consistent / uniform (e.g. Figure 3B labelling of figures vs. Figure legend)*

- Lines 136 and 139: 'hereafter called the KKE/D domain' is duplicated
- Table S2: nomenclature of double mutants not uniform (e.g. *cbf5-Δkk* versus *cbf5-ΔΔk* versus)
- line 330: check punctuation (we suspect there should be a comma)
- line 309: please spell out what the *cbf5-Δkk* mutant is somewhere in the text

We thank Reviewer #3 for pointing out these mistakes and omissions. We have corrected each point accordingly in the revised manuscript.

(14) Figure 3B and others: red / green is not ideal for color-blind people

We modified all microscopy panels in which there was no individual image for each color in the initial version of manuscript to facilitate color visualization for color-blind people. Moreover, the overlaps can be clearly distinguished as they are in yellow and average plots also help to distinguish the signal profiles of mKate/mCherry and GFP in the nucleus. We also would like to highlight that Net1 or Nop1 signals are mainly used as spatial controls allowing to show that our Z-plan showing the GFP signal is in the nucleolus. Consequently, the merged images are not critical to support our claim. We nevertheless provided a large panel with modified colors in the case of **Supplemental Figure S6B**, as we specifically commented the adjacent localization between snoRNPs and Net1 in the main text.

(16) Figure 3E: the methods section for the qPCR is missing (experimental protocol, quantification, normalization). Therefore, it is very hard to understand and judge how the scale (IP/input) in 3E was calculated.

We apologize for this omission and have added a specific "qPCR analysis" paragraph in the Methods section, **Line 761**.

(17) Figure 3F: please add arrows / labels and / or a schematic representation to guide the reader what to look at

As discussed with Reviewer #2, we added a new figure (**New Supplemental Figure S3F**) and we labelled the **Figure 3F** to help the reader understand this panel.

(18) Figure 3A / 4A: please clarify the schemes; the KKE / KKE/D domain seems to be differently defined (e.g. the arrows Figure 3 vs Figure 4 (402-483 vs 392-483), and clarify what was used in which experiment.

As discussed in point (4), we have tried to clarify the rationale between these different constructions and have modified **Figures 3A/4A** and their legends accordingly.

(19) S6B, pictures of post-diauxic stress are not mentioned or used in the paper

We mentioned the post-diauxic stress in the new version of the manuscript (**Line 341**).

RESPONSE TO REVIEWERS' COMMENTS

Reviewer #1:

In my initial review I noted that I felt the work would be highly significant to the field, as the contribution of LLPS to nucleolar formation and spatial partitioning is hotly debated. That said, there were several issues raised by me and the other reviewers. I feel that the revised manuscript has been strengthened by data added to address these comments, including additional CLEM images, assessment of additional nucleolar factors and demonstration of the statistical significance of the growth assay and Northern blot results. The text revisions, which addressed both minor (grammar, consistency) and major (clarification of certain methods and results) issues, have also improved readability.

Reviewer #2:

This revised manuscript includes detailed responses to the many questions posed by the reviewers. In that regard, the job is done; however, neither the title nor the abstract announces one key take-home message and its significance. If the authors can modify these items they will heighten the impact of their findings.

We agree with Reviewer #2 that mentioning the highly conserved eukaryotic ribonucleoprotein particles "snoRNPs" in the title could increase the visibility of the article. We modified the title slightly as follows: "The dual life of disordered lysine-rich domains of snoRNPs in rRNA modification and nucleolar compaction."

Reviewer #3:

The authors have addressed my comments and put together a beautiful manuscript.

We thank Reviewer #3 for his very positive feedback.

There is only one point / figure that needs to be clarified further, which is I I/J. For condensates in and higher salt, the authors alter in the description between "sensitive to the addition" and "in the presence of" (the latter being more likely when reading the methods part). It is unfortunately important whether the condensates were formed in the presence of high salt, or whether they were first formed in low salt and then dissolved by addition of 300 mM salt, which would demonstrate reversibility, distinguishing them from aggregates, and thus what I would suggest should be the goal of this experimental setup. Therefore, I would suggest to repeat this experiment accordingly (i.e. first make condensates, then add high salt).

We understand Reviewer #3's comment regarding the potential reversibility of condensate formation. As requested, we added data showing that KKE/D domain/RNA coacervates formed in the absence of salt are no longer stable after addition of 300 mM NaCl (**Supplemental Figure 1F**, see below). We presented and commented these results (**Lines 153-161**) in our revised manuscript without going into too many details due to length limitations, also because further experiments beyond the scope of this study will be required to characterize the specific properties of these condensates as well as the exact dissolution kinetics of these coacervates. Nevertheless, we hope that our work paves the way for future *in vitro* studies.

New Supplemental Figure 1F:

(F) Coacervates were formed *in vitro* by incubating for 15 min at room temperature a mix of [KKE/D]x9-Alexa 488 peptide and either poly-uridine (poly-U) RNAs or total RNA from *S. cerevisiae*. Coacervate suspension was next observed 60 min following addition of a Tris-buffered solution containing NaCl (+ NaCl, 300 mM final NaCl concentration) or the Tris-buffered solution without NaCl (+ Veh). Scale bar = 10 μ m.

Furthermore, there are some aggregates visible in 1J high salt in the brightfield, but not in the GFP channel - could the authors please briefly explain this discrepancy?

These structures visible in the bright field channel in Figure 1J or Supplemental Figure 1F are not observable in the suspension containing only the KKE/D domain conjugated to Alexa 488 (Figure 1I) and are formed only due to the presence of non-denatured total RNA. They are therefore visible independently of the ability of KKE/D domains to form condensates around these large RNA structures. This last point explains why we still observe these structures in the brightfield channel in the absence of KKE/D domain condensates (in the presence of salt). We have clarified and rephrased this part of the text (Lines 153-161).